# Generalize or Detect? Towards Robust Semantic Segmentation Under Multiple Distribution Shifts

Zhitong Gao[1,2]    Bingnan Li[1]    Mathieu Salzmann[2]    Xuming He[1,3]

[1]ShanghaiTech University    [2] EPFL

[3]Shanghai Engineering Research Center of Intelligent Vision and Imaging

`{gaozht, libn, hexm}@shanghaitech.edu.cn`

`mathieu.salzmann@epfl.ch`

## Abstract

In open-world scenarios, where both novel classes and domains may exist, an ideal segmentation model should detect anomaly classes for safety and generalize to new domains. However, existing methods often struggle to distinguish between domain-level and semantic-level distribution shifts, leading to poor out-of-distribution (OOD) detection or domain generalization performance. In this work, we aim to equip the model to generalize effectively to covariate-shift regions while precisely identifying semantic-shift regions. To achieve this, we design a novel generative augmentation method to produce coherent images that incorporate both anomaly (or novel) objects and various covariate shifts at both image and object levels. Furthermore, we introduce a training strategy that recalibrates uncertainty specifically for semantic shifts and enhances the feature extractor to align features associated with domain shifts. We validate the effectiveness of our method across benchmarks featuring both semantic and domain shifts. Our method achieves state-of-the-art performance across all benchmarks for both OOD detection and domain generalization. Code is available at `https://github.com/gaozhitong/MultiShiftSeg`.

## 1   Introduction

Semantic segmentation, a fundamental task in computer vision, has become indispensable in various real-world applications, such as autonomous driving [35]. Recent progress in deep learning-based semantic segmentation has exhibited promising results under the assumption of consistent distributions between the training and testing data. However, these models often falter when faced with distributional shifts. Consequently, research on semantic segmentation under distributional shifts has garnered significant attention in recent years. Some studies approach this challenge from a generalization perspective, aiming to train networks to adapt to data with covariate distribution shifts, such as novel domains [9, 45]. Another line of research focuses on training models to discern (or detect) test data exhibiting semantic distributional shifts, such as anomalies or unfamiliar objects, to ensure reliable predictions [5, 4]. In real-world situation, both types of distribution shifts often occur jointly. This leaves us with the question: *Can a model jointly handle both kinds of distribution shift?*

To address this question, we assess the ability of current domain generalization techniques [9, 45] to detect unknown objects and that of out-of-distribution detection techniques [31, 42, 46] to generalize to unknown domains. Interestingly, we find that models trained using domain generalization techniques, such as domain randomization or whitening transformation, often fail to identify unknown objects, and sometimes even perform worse than the baseline without domain generalization. Furthermore, we observe that models trained using out-of-distribution detection techniques struggle to generalize to unknown domains, exhibiting overly high uncertainty towards objects experiencing domain shifts compared to baseline methods without OOD training. While one intuitive approach is

38th Conference on Neural Information Processing Systems (NeurIPS 2024).

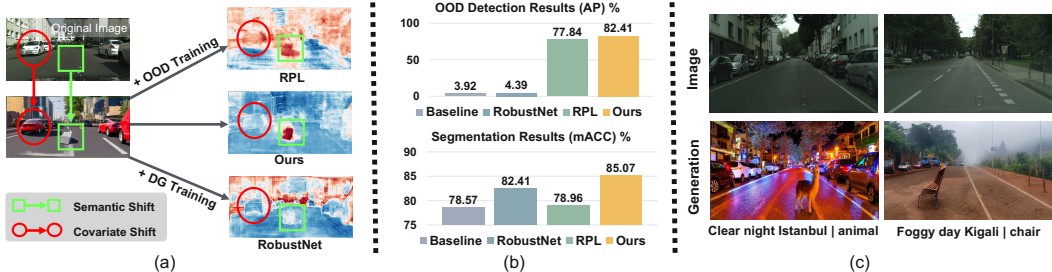

Figure 1: We study semantic segmentation with both **semantic-shift** and **covariate-shift** regions. (a) Training for *Out-of-distribution (OOD) detection* alone [31] yields high uncertainty for both types of shifts, whereas training for *domain generalization (DG)* alone [9] tends to produce low uncertainty for both. Our method effectively differentiates between the two, generating high uncertainty only for semantic-shift regions. (b) We achieve strong performance in both OOD detection and domain-generalized semantic segmentation. (c) This is achieved by coherently augmenting original images (first row) with both covariate and semantic shifts (second row).

to combine existing anomaly segmentation and domain generalization techniques during training, we note that current domain generalization strategies primarily address image-level shifts, whereas anomaly segmentation focuses on object-level semantic differences. Consequently, the resulting models tend to generalize well to image-level variations, such as changes in weather but struggle with object-level shifts. They often misinterpret any object-level distribution shift as a semantic anomaly, assigning high uncertainty scores to known objects that exhibit covariate changes, such as color variations in cars or changes in pedestrian attire, as demonstrated in Fig. 1. These experiments underscore the challenge of differentiating and jointly handling different types of distribution shifts.

In this work, we jointly study both semantic and covariate distribution shifts. That is, we aim to equip the model to generalize effectively to covariate-shift regions while precisely identifying semantic-shift regions. To achieve this, we design a novel generative augmentation method to produce coherent images that incorporate both anomaly (or novel) objects and various covariate shifts at both image and object levels. Furthermore, we introduce a training strategy that re-calibrates uncertainty specifically for semantic shifts and enhances the feature extractor to align features associated with domain shifts [1].

Specifically, we first introduce a novel data augmentation technique that employs a semantic-to-image generation model to create data that encompasses both covariate and semantic shifts at various levels, allowing the model to learn the essential differences between the shift types. Additionally, we introduce a learnable, semantic-exclusive uncertainty function trained using a relative contrastive loss. We adopt a two-stage training paradigm designed to balance the integration of these enhancements while minimizing their potential interference. A noise-aware training strategy further complements this approach, employing online, pixel-wise selection to mitigate noise in the generated images. Altogether, our approach not only boosts the model's generalization across domain shifts but also ensures a high level of uncertainty in response to semantic shifts.

We validate the effectiveness of our method across benchmarks featuring both semantic and domain shifts, including RoadAnomaly [30], SMIYC [5], ACDC-POC [12] and MUAD [15] benchmarks. Our results demonstrate that our method achieves state-of-the-art performance across all benchmarks, employing different segmentation backbones for both OOD detection and known class segmentation.

In summary, our contributions are: (1) We study semantic segmentation under both semantic and domain shifts, revealing limitations in methods focused on a single shift; (2) We introduce a coherent-generative augmentation method that augments training data with both shifts; (3) We propose a two-stage, noise-aware training pipeline to optimally leverage augmented data, learning a semantic-exclusive uncertainty function while aligning features for domain shifts.

## 2 Related Work

**Anomaly Segmentation**   (a.k.a. dense out-of-distribution detection). The task aims to detect anomalies or unknown objects by producing pixel-level uncertainty maps. One approach uses generative

---

[1]In this work, we use 'domain shift' and 'covariate shift' interchangeably.

models to learn training distributions and detect anomalies through reconstruction differences [30, 48], but this often requires additional networks, resulting in slower inference. Other methods use auxiliary OOD data to train models to distinguish known from unknown instances [46, 6, 31, 42, 36, 18]. Among these, Entropy Maximization [6] uses entire images from COCO [29] as OOD proxies, maximizing softmax entropy on these samples. PEBAL [46] improves upon this by cutting out OOD object instances, pasting them into training images, and using an energy function as the uncertainty score. To reduce artifacts in the pasted OOD region, [52] proposes using a style transfer model to align the pasted region with the background. RPL [31] further regularizes embedding similarity between COCO background pixels and training images. Beyond improvements in OOD proxies and uncertainty functions, recent methods explore the use of the Mask2Former architecture [8], such as RbA [36], Mask2Anomaly [42], and EAM [18]. Our method follows this second approach, generating OOD data with a semantic-to-image model to reduce artifacts and further introducing a learnable uncertainty function to enhance both OOD detection and known class segmentation. It is architecture-agnostic, compatible with both pixel-based and mask-based segmentation backbones.

**Domain Generalization for Semantic Segmentation** The task aims to train a model on one or more source domains that can perform well on unseen target domains. Existing techniques focus either on introducing specialized model architectures, such as those incorporating normalization [39] or whitening transformations [9, 41, 27], or on designing domain randomization techniques [45, 50, 23]. Most domain randomization methods rely on image transformation rules or style transfer [45, 50]. Recently, Jia et al. [23] proposed a semantic-to-image model that generates images across diverse domains. Orthogonally, Bi et al. [3] explore architectural changes with Mask2Former. Our approach belongs to the domain randomization category, generating images with both domain and semantic shifts simultaneously to improve the model's ability to distinguish between these shifts.

**Segmentation Under Multiple Distribution Shifts** Early works [51, 2] demonstrated the necessity and feasibility of addressing both semantic segmentation under domain shifts and anomaly segmentation. However, these problem settings remain in their early stages (e.g., image-level anomalies) and may not fully capture the true challenges. More recent benchmarks, such as RoadAnomaly [30], SMIYC [5], and MUAD [15], include domain and semantic shifts that better reflect real-world scenarios. Some recent studies [16] have explored the effects of domain shifts on anomaly segmentation benchmarks and proposed a test-time adaptation pipeline to address the problem. In this work, we aim to further bridge the gap by investigating the core challenges of adopting domain generalization techniques and simultaneously enhancing model performance in both areas.

**Generative-based Data Augmentation** This technique is widely used to expand training datasets and prevent overfitting [14, 38, 20, 23, 12]. Unlike rule-based augmentation methods, which focus on image-level changes, generative methods can introduce more object-level variations. Among these works, [12, 33] are the most related to ours, using a text-guided inpainting pipeline to generate anomalies or novel objects. However, this local generation process risks creating inconsistencies between the patch and its background. Additionally, they either focus on generating novel objects within the same domain [12] or use separate pipelines for domain and semantic shifts [33]. In contrast, our method generates multiple distribution shifts in a single process, preserving the global context of the image and ensuring a more natural integration of novel objects.

## 3 Method

### 3.1 Problem Formulation and Method Overview

We consider the problem of *semantic segmentation under multiple distribution shifts*. Formally, we define the training distribution as $P_{XY}$ in $\mathcal{X} \times \mathcal{Y}_{in}^{H \times W}$, where $\mathcal{X} = \mathbb{R}^{3 \times H \times W}$ represents the three-dimensional input space of images with $H \times W$ pixels, and $\mathcal{Y}_{in} = [1, C]$ denotes the semantic label space at each pixel. The test distribution is denoted as $Q_{XY} \in \mathcal{X} \times \mathcal{Y}_{test}^{H \times W}$. There are two common types of distribution shifts: *covariate shifts*—where the input distribution changes ($Q_X \neq P_X$) but the label space remains the same —and *semantic shifts*, which involve alterations to the label space, including the introduction of novel categories ($\mathcal{Y}_{test} \neq \mathcal{Y}_{in}$). We consider the possibility of both types of distribution shifts occurring during testing.

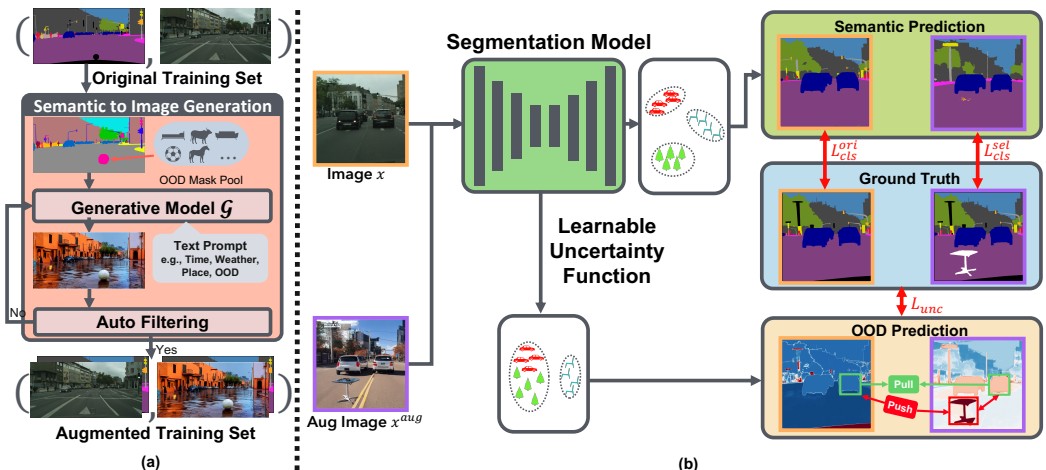

Figure 2: Method Overview: (a) A novel generative-based data augmentation strategy that supplements training data with both covariate and semantic shifts in a coherent manner. (2) A semantic-exclusive uncertainty function with two-stage noise-aware training to encourage invariant feature learning for covariate-shift regions while maintaining high uncertainty for semantic-shift regions.

Our goal is to learn a model capable of jointly identifying semantic-shift regions and generalizing well under covariate shifts. This involves two primary challenges: (1) Enabling the model to distinguish between the two types of distribution shifts, and (2) ensuring the model responds appropriately to each type. To address the first challenge, we introduce a novel generative-based data augmentation strategy that supplements training data with both covariate and semantic shifts in a coherent manner. To tackle the second challenge, we propose a semantic-exclusive uncertainty function with a decoupled training strategy. This encourages the model to learn invariant features for covariate-shift regions while maintaining high uncertainty for semantic-shift regions. Below, we first introduce our generative-based augmentation strategy (Section 3.2), followed by the model training pipeline (Section 3.3).

## 3.2 Coherent Generative-based Augmentation

To distinguish between covariate and semantic shifts, we design a coherent generative-based data augmentation (CG-Aug) pipeline that enriches the training data with realistic and diverse distribution shifts. The pipeline consists of two stages: The first stage uses zero-shot semantic-to-image generation to create a variety of synthetic data, while the second stage automatically filters out low-quality synthetic data. We describe the details of each stage below.

**Zero-Shot Semantic-to-Image Generation.** To generate more realistic and diverse OOD data for segmentation, we propose a generation process that first cut-and-pastes the semantic mask of novel objects to the training labels and then leverages a semantic-to-image generation model to create corresponding augmentation images. By exploiting powerful image generation models, this process is able to produce images with a wide range of covariate shifts and augments the training images with both covariate and semantic shifts in a coherent manner. We detail our process below.

Formally, given training set $\mathcal{D}^{tr} \coloneqq \{(x_n, y_n)\}_{n=1}^{N_t}$ with $(x_n, y_n) \sim P_{XY}$, we introduce an auxiliary OOD set $\mathcal{D}^o \coloneqq \{y_m^o\}_{m=1}^{N_a}$ with object masks $y_m^o \in \mathcal{Y}_{out}^{H \times W}$. Subsequently, using a pretrained semantic-to-image generative model $\mathcal{G} : (\mathcal{Y}_{in} \cup \mathcal{Y}_{out})^{H \times W} \to \mathbb{R}^{H \times W}$, we generate an augmented image as

$$x^{aug} = \mathcal{G}(y^{aug}, t) \qquad \text{with} \quad y^{aug} = y \oplus y^o, \tag{1}$$

where $t$ is a text prompt and $\oplus$ denotes the pasting operation. Here we adopt a pretrained Control-Net [53] as $\mathcal{G}$ to instantiate the semantic-to-image generation process. Thanks to the powerful prior encoded in Stable Diffusion [43], this process allows us to generate images with more diverse styles than a task-specific semantic-to-image generation model, therefore creating rich covariate shifts. Moreover, we leverage the text prompt $t$ to produce more diversity in the augmented images by specifying the space, time, and weather, and to enhance the OOD object generation by indicating the class of the pasted objects, via a set of templates (see Appendix A.1 for details).

**Auto-Filtering.** While generative-based augmentation can produce more diverse and realistic distribution shifts than rule-based augmentation, we observed it to often yield inaccurate or noisy rendering for the OOD objects. This might be caused by the fact that these objects appear rarely, or their cut masks are inconsistent with the surroundings. To cope with this, we design an automatic filtering process that identifies generation failures where no object is generated, or a known-category object is incorrectly generated. To achieve this, we leverage pretrained segmentation models to check the region size or its semantic class, and assign a quality score to each generated image. We then filter out the images with low-quality scores (see Appendix A.2 for details).

We perform the above image generation process offline before model training, and the resulting synthetic data is used alongside standard augmentation strategies, such as mixup and AnomalyMix [46], during training. Below, we denote our augmented dataset as $D^{aug} = \{(x_n, y_n, x_n^{aug}, y_n^{aug})\}_{n=1}^{N_t}$.

### 3.3 Model Training

Given the augmented dataset, we aim to train a segmentation model with recalibrated uncertainty output, generating high OOD scores for semantic-shift regions and performing robustly under covariate shifts. To achieve this, we propose a learnable uncertainty function and develop a stage-wise learning strategy that initializes the uncertainty function before fine-tuning the entire model. Our training process integrates a relative contrastive loss and a noise-aware data selection scheme, enabling the model to effectively align both the feature space and the OOD output scores. We note that our method is generic and can be applied to pixel-wise models (e.g. DeepLabv3+ [7]) or mask-wise models (e.g. Mask2Former [8]).

#### 3.3.1 Semantic-Exclusive Uncertainty Recalibration

**Learnable Uncertainty Function.** Suppose we have a neural network with its feature extractor $f(x) \in \mathbb{R}^{M \times F}$, where $M$ is the number of pixels (or masks), and $F$ is the feature dimension. We introduce a learnable linear projection $W^o \in \mathbb{R}^{F \times C}$, with $W_c^o$ denotes $W^o[:c]$ for short. For a pixel-wise prediction model, we adopt the energy function form and parameterize it into a learnable uncertainty function

$$u(x) = \log \sum_c \exp f(x) W_c^o. \tag{2}$$

For a mask-based segmentation network, we use the adapted maximum softmax probability (MSP) defined in [8] as the uncertainty function, and parameterize it with $W^o$, leading to

$$u(x) = \max_c \left( \text{softmax} \left( f(x) W_c^o \right)^T \cdot g(x) \right). \tag{3}$$

Here $g(x) \in (0,1)^{M \times H \times W}$ is the sigmoid output of the mask head. For both cases, we initialize the projection function $W^o$ as the class weight $W^{in}$ of the pretrained segmentation network. The corresponding uncertainty score corresponds to the original energy score (or MSP score).

**Relative Contrastive Loss.** We train the uncertainty function using a novel relative contrastive loss, which encourages higher uncertainty in unknown-class regions compared to known-class regions, while ensuring that regions with and without covariate shifts exhibit similar levels of uncertainty.

Formally, for each batch of data $\{(x_n, y_n, x_n^{\text{aug}}, y_n^{\text{aug}})\}_{n=1}^B$, where $B$ is the batch size, we define the following pixel index sets: $\Omega^{\text{in}} = \{i : y(i) \in \mathcal{Y}_{\text{in}}\}$, representing *inlier* pixel indices from the original training images; $\Omega^{\text{aug}} = \{i : y^{\text{aug}}(i) \in \mathcal{Y}_{\text{in}}\}$, representing inlier pixel indices from the *augmented* training images (covariate-shift set); and $\Omega^{\text{out}} = \{i : y(i) \notin \mathcal{Y}_{\text{in}}\} \cup \{i : y^{\text{aug}}(i) \notin \mathcal{Y}_{\text{in}}\}$, representing *outlier* pixel indices from both original and augmented images (semantic-shift set). Here, $y(i)$ (or $y^{\text{aug}}(i)$) denotes the label of pixel $i$. Our contrastive loss is defined as

$$L_{\text{unc}} = \sum_{o \in \Omega^{\text{out}}, i \in \Omega^{\text{in}}} \tau_{\lambda_1}(u_o - u_i) + \sum_{o \in \Omega^{\text{out}}, c \in \Omega^{\text{aug}}} \tau_{\lambda_2}(u_o - u_c) + \sum_{c \in \Omega^{\text{aug}}, i \in \Omega^{\text{in}}} m_{c,i} \cdot \tau_{\lambda_3}(-(u_c - u_i)), \tag{4}$$

where $\tau_\lambda(x) = \max(\lambda - x, 0)$ is the margin-based contrastive loss, which encourages the input value to exceed $\lambda$. The first two terms promote larger uncertainty gaps between unknown-class and known-class regions, while the third term encourages smaller uncertainty gaps between covariate-shifted and

original data. For the third term, we calculate gaps only between pairs of original and augmented images, with $m_{c,i} \in \{0, 1\}$ indicating whether pixel $(c, i)$ is paired in the dataset. The three margin values $(\lambda_1, \lambda_2, \lambda_3)$ introduce priors on the uncertainty gaps, and are set based on the initial average distance. Our method remains robust across a wide range of margin values (cf. Table 6).

Compared to existing OOD losses that either maximize uncertainty only for unknown data [31] or supervise known and unknown data separately [46, 42], our loss supervises the relative distance between them, making it more robust to hyperparameters and simpler to train (cf. Sec.4.5).

### 3.3.2 Two-Stage Noise-Aware Training

We now present our two-stage training procedure, which sequentially learns the uncertainty function and the feature extractor $f(x)$ of the segmentation network. Specifically, we first freeze the pre-trained segmentation network and learn the semantic-exclusive uncertainty function using the relative contrastive loss defined in Eq. 4. We then fine-tune the feature extractor with both the contrastive and standard segmentation loss to improve the feature representations of both known and OOD classes.

Despite the offline filtering process, the generated images may still contain regions that are inconsistent with the label masks. To address this, we introduce a pixel-wise sample selection scheme during training, based on the 'small loss' criterion [1]. Specifically, we compute and rank the cross-entropy loss for each pixel, selecting pixels with smaller losses for backpropagation while ignoring those with larger losses. Formally, our *selective cross-entropy loss* is defined as

$$L_{\text{seg}}(y, p, \eta) = \sum_i \eta_i \sum_c y_i^c \log p_i^c \, , \tag{5}$$

where $p_i$ and $y_i$ represent the pixel-wise softmax score and one-hot label, respectively, and $\eta_i \in \{0, 1\}$ indicates whether a pixel is selected for backpropagation. We determine the percentage of selected pixels per batch by visualizing the selection map of a small number of samples, ensuring that visibly incorrect patterns are excluded (see Figure 4 for an example). For models using the Dice loss, such as mask-based ones, we use a similar scheme to remove pixels with a large loss (cf. Appendix A.3).

For the original data, which we assume to be noise-free, we set $\eta_i = 1$ for all pixels. This corresponds to using the *standard cross-entropy loss*. We denote the segmentation loss for the original data as $L_{\text{seg}}^{\text{in}}$ and for the generated augmentation data as $L_{\text{seg}}^{\text{aug}}$. The overall loss function can be written as

$$L = L_{\text{unc}} + \beta_1 L_{\text{seg}}^{\text{in}} + \beta_2 L_{\text{seg}}^{\text{aug}}. \tag{6}$$

Here, $\beta_1$ and $\beta_2$ ensure that the three loss terms are on the same scale.

In summary, our semantic-exclusive uncertainty function, trained through a decoupled parameter training approach and relative contrastive loss, enables the model to fully leverage the generated distribution-shift data. Our noise-aware learning strategy enhances the model's robustness against generation errors. Together, these components of our training pipeline equip the model to effectively learn both domain generalization and accurate OOD detection, ensuring robust performance in dynamic open-world scenarios.

## 4 Experiments

In this section, we evaluate our method's performance in jointly handling anomaly segmentation and domain generalization using several datasets that include both domain and semantic shifts: RoadAnomaly [30], SMIYC [5], ACDC-POC [12], and MUAD [15]. We first introduce the datasets in Sec.4.1 and describe the experimental setup in Sec.4.2. The results are presented in Sec.4.3 and Sec.4.4, followed by an ablation study in Sec. 4.5.

### 4.1 Datasets

Following the literature [5, 31, 8], we train our model on the Cityscapes dataset [11] and evaluate its performance on the test sets described below. Based on the evaluation goals, we divide these datasets into two groups. Examples from each dataset are shown in Fig. 3.

**Anomaly Segmentation Datasets:** (a) *The Road Anomaly dataset* [30] includes 60 images of real-world road anomalies such as animals, rocks, and obstacles, featuring various driving conditions

Table 1: **Results on anomaly segmentation benchmarks:** RoadAnomaly, SMIYC-RA21 and SMIYC-RO21. Our method achieves the best results under both backbones (Best results in Bold).

| Method | Backbone | RoadAnomaly | | | SMIYC - RA21 | | SMIYC - RO21 | |
|---|---|---|---|---|---|---|---|---|
| | | AUC ↑ | AP ↑ | FPR$_{95}$ ↓ | AP↑ | FPR$_{95}$ ↓ | AP ↑ | FPR$_{95}$ ↓ |
| Maximum softmax [21] | | 67.53 | 15.72 | 71.38 | 27.97 | 72.05 | 15.72 | 16.60 |
| ODIN [28] | | - | - | - | 33.06 | 71.68 | 22.12 | 15.28 |
| Mahalanobis [26] | | 62.85 | 14.37 | 81.09 | 20.04 | 86.99 | 20.90 | 13.08 |
| Image resynthesis [30] | | - | - | - | 52.28 | 25.93 | 37.71 | 4.70 |
| SynBoost [13] | | 81.91 | 38.21 | 64.75 | 56.44 | 61.86 | 71.34 | 3.15 |
| Maximized entropy [6] | DeepLabv3+ | - | 48.85 | 31.77 | 85.47 | 15.00 | 85.07 | 0.75 |
| PEBAL [46] | | 87.63 | 45.10 | 44.58 | 49.14 | 40.82 | 4.98 | 12.68 |
| Dense Hybrid [17] | | - | 31.39 | 63.97 | 77.96 | 9.81 | 87.08 | **0.24** |
| RPL+CoroCL [31] | | 95.72 | 71.61 | 17.74 | 83.49 | 11.68 | 85.93 | 0.58 |
| Ours | | **96.40** | **74.60** | **16.08** | **88.06** | **8.21** | **90.71** | 0.26 |
| Mask2Anomaly [42] | | - | 79.70 | 13.45 | 88.7 | 14.60 | 93.3 | 0.20 |
| RbA [36] | Mask2Former | - | 85.42 | 6.92 | 90.90 | 11.60 | 91.80 | 0.50 |
| M2F-EAM [18] | | - | 69.40 | 7.70 | **93.75** | **4.09** | 92.87 | 0.52 |
| Ours | | **97.94** | **90.17** | **7.54** | 91.92 | 7.94 | **95.29** | **0.07** |

and covariate shifts. (b) *The SMIYC benchmark* [5] consists of RoadAnomaly21 (10 validation, 100 test images) and RoadObstacle21 (30 validation, 327 test images), with anomaly objects and domain shifts. These datasets provide masks for anomaly objects, allowing us to evaluate our method's performance on anomaly segmentation under distribution shifts.

**Joint Anomaly Segmentation and Domain Generalization Datasets:** (a) The *ACDC-POC dataset* [12] is based on the original ACDC Validation set [44] with generated anomaly objects via inpainting [12]. It contains 200 images with domain shifts including various weather and night scenes. (b) The *MUAD dataset* [15] is a synthetic dataset containing various driving environments and anomaly objects. We use the challenge test set as in [49], which contains 240 images with domain shifts at both object and image levels, and anomaly objects such as animals and trash cans.[2] These two datasets contain both known-class annotations and unknown object masks, enabling us to evaluate our method jointly for anomaly segmentation and domain generalization.

## 4.2 Experimental Setup

**Performance Measure:** For evaluation of anomaly segmentation, we use the Area Under the Receiver Operating Characteristics curve (AUROC), the Average Precision (AP), and the False Positive Rate at a True Positive Rate of 95% (FPR95). For evaluation of known class segmentation, we use the mean intersection-over-union (mIoU) and the mean accuracy (mAcc).

**Implementation Details:** We build our method on two segmentation backbones: (a) DeepLabv3+ [7] and (b) Mask2Former [8]. We maintain the network architecture, pretrained models, segmentation loss, and training pipeline the same as in previous work [31, 42] to make a fair comparison. We use the SMIYC validation set for model selection and maintain the same model for evaluation across all test sets. We refer the reader to Appendix A for other training details.

## 4.3 Results on Anomaly Segmentation Benchmarks

We present the performance of our method on anomaly segmentation benchmarks, including Road-Anomaly and SMIYC (RA21 and RO21). As shown in Table 1, our method achieves state-of-the-art performance on both DeepLabv3+ and Mask2Former-based models. With the same backbone, it outperforms RPL [31] by 3% on RoadAnomaly and 5% on SMIYC, and surpasses Mask2Anomaly [42] by 10% on RoadAnomaly and 3% on SMIYC. Recent methods, M2F-EAM [18] and RbA [36], use a more powerful Swin Transformer backbone, while ours uses ResNet-50, as Mask2Anomaly. M2F-EAM also uses Mapillary Vistas [37] as additional dataset for training. Despite these unfair comparisons, our method still outperforms both on most metrics, demonstrating its effectiveness.

---

[2]Since we use Cityscapes as the training set, the unknown object set differs from that used in [49].

Table 2: **Results on ACDC-POC and MUAD**. Our model achieves the best performance in both anomaly segmentation (AP↑ , FPR↓ ) and domain-generalized segmentation (mIoU↑ , mAcc↑ ). Anomaly segmentation methods typically perform worse than the baseline for known class segmentation, while domain generalization methods fall below the baseline on OOD detection. (Best results are in bold; results below baseline are in blue.)

| Method | Backbone | OOD | DG | AP↑ | FPR$_{95}$ ↓ | mIoU↑ | mAcc↑ | AP↑ | FPR$_{95}$ ↓ | mIoU↑ | mAcc↑ |
|---|---|---|---|---|---|---|---|---|---|---|---|
| | | | | | ACDC-POC | | | | MUAD | | |
| Baseline [7] | | - | - | 3.92 | 55.50 | 46.89 | 78.57 | 1.34 | 72.78 | 29.47 | 68.63 |
| RuleAug [45] | | - | ✓ | 2.09 | 72.79 | 48.60 | 81.79 | 0.99 | 81.08 | 29.42 | 69.22 |
| RobustNet [9] | | - | ✓ | 4.39 | 62.65 | 47.41 | 82.41 | 2.27 | 58.64 | **32.18** | 72.02 |
| PEBAL [46] | DeepLabv3+ | ✓ | - | 20.67 | 14.35 | 45.59 | 81.28 | 7.81 | 47.56 | 29.08 | 66.41 |
| RPL [31] | | ✓ | ✓ | 77.84 | 1.20 | 46.35 | 78.96 | 27.70 | 24.45 | 29.86 | 71.60 |
| OOD + RuleAug [45] | | ✓ | ✓ | 80.65 | 1.30 | 46.76 | 73.08 | 20.97 | 20.37 | 27.83 | 63.02 |
| Ours | | ✓ | ✓ | **82.41** | **1.01** | **54.12** | **85.07** | **36.08** | **18.74** | 31.33 | **73.13** |
| Mask2Anomaly [42] | | ✓ | - | 73.77 | 3.60 | 47.32 | 83.10 | 39.32 | 41.24 | 23.43 | 61.91 |
| OOD + RuleAug [45] | Mask2Former | ✓ | ✓ | 82.82 | 0.79 | 50.36 | 82.83 | 25.43 | 41.15 | 26.27 | 67.51 |
| Ours | | ✓ | ✓ | **90.42** | **0.46** | **51.75** | **83.16** | **45.65** | **24.70** | **28.44** | **73.77** |

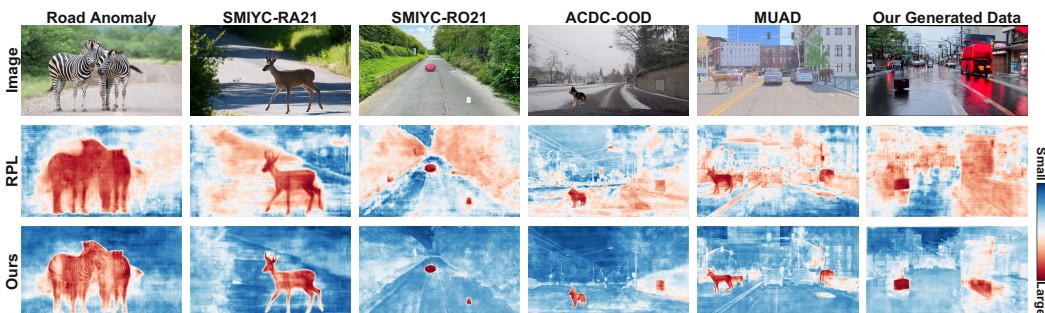

Figure 3: **Comparison of Uncertainty Maps.** Our method robustly detects anomalies under covariate shifts across five datasets (first five columns) and generated data (last column). The previous method RPL [31] failed to distinguish domain from semantic shifts, producing high uncertainty in both cases.

In Fig. 3, we visualize the uncertainty map output by our method using the DeepLabv3+ architecture. Compared to the previous state-of-the-art method, RPL [31], our model assigns higher uncertainty scores to anomalous objects and lower uncertainty scores to covariate shifts. This highlights the efficacy of our method in distinguishing between domain shifts and semantic shifts.

## 4.4   Results on ACDC-POC and MUAD

We then extend our evaluation to the ACDC-POC and MUAD datasets, assessing both anomaly segmentation performance and known-class domain generalization performance. For a comprehensive comparison, we include both previous state-of-the-art OOD detection techniques and domain generalization techniques [9, 45]. Additionally, we trained a DG+OOD combination method by combining naive OOD training with contrastive loss and rule-based data augmentation (denoted as OOD+RuleAug). A DeepLabv3+ model with standard training is used as a baseline method. [3]

The results are shown in Table 2, where our model achieves the best results for both out-of-distribution detection and domain generalization, demonstrating its capacity in jointly handling both types of distribution shifts. By comparison, previous methods fall short in either known class segmentation or OOD detection. Specifically, we find that: (a) Previous works that mainly focus on **domain generalization** (RobustNet [9], RuleAug [45]) generally improve the known class segmentation results, but their performance in OOD detection is affected, sometimes worse than the baseline. (b) Previous works that mainly focus on **OOD detection** (such as PEBAL [46]) perform poorly

---

[3] For all compared methods, except RuleAug, we used the official pretrained models provided by the respective authors and performed re-inference to obtain the results. For RuleAug, we applied a combination of color jittering, Gaussian blur, etc., as suggested in [45]. For further details, please refer to the Appendix A.5

Table 3: **Impact of CG-Aug and Training Strategy.** The proposed coherent generative-based augmentation consistently enhances the previous OOD method, Mask2Anomaly [42] (M2A for short). Our fine-tuning strategy makes better use of the data and further boosts the performance.

| | | RoadAnomaly | | SMIYC-RA Val | | SMIYC-RO Val | |
|---|---|---|---|---|---|---|---|
| Training | Aug. | AP↑ | FPR$_{95}$ ↓ | AP↑ | FPR$_{95}$ ↓ | AP↑ | FPR$_{95}$ ↓ |
| M2A [42] | Default | 79.70 | 13.45 | 94.50 | 3.30 | 88.60 | 0.30 |
| M2A [42] | Ours | 85.47 | 22.38 | **97.96** | 1.55 | 89.80 | **0.12** |
| Ours | Ours | **90.17** | **7.54** | 97.31 | **1.04** | **93.24** | 0.14 |

on domain generalization, sometimes worse than the baseline. Furthermore, their OOD detection performance may also be affected by the domain shift. (c) Previous works that jointly handle **image-level DG and OOD** (RPL [31] and OOD+RuleAug) may not fully distinguish object-level domain shifts. Our method leveraging diverse augmentations and a dedicated decoupled training strategy enables the model to jointly handle OOD detection and domain generalization.

In Appendix C.1, we provide additional results on individual domain shifts (fog, rain, snow, night) and per-class evaluation. Furthermore, we compare our method with other DG methods on the original ACDC dataset in Appendix C.4, where we show superior domain generalization performance.

## 4.5    Analysis and Ablation Study

We conduct ablation studies to evaluate the design of our components. We begin by analyzing the effectiveness of our proposed modules: the coherent generative-based augmentation (CG-Aug) and our model training strategy. We then proceed with a detailed examination of each module's design.

**Impact of CG-Aug and Training Strategy**    We evaluate the decoupled contributions of our data augmentation and training strategies in Table 3. Starting with a recent anomaly segmentation method, Mask2Anomaly [42], we first replace its original OOD data, which utilizes cut-and-pasted COCO images, with our proposed CG-Aug. As shown in Row #2, this substitution results in consistent performance improvements across all datasets. This demonstrates the efficacy of introducing data with both semantic and domain shifts in a coherent way. Next, we replace their training strategy with ours, leading to further gains in performance. This indicates that our training strategy is more effective in leveraging the generated data. Additionally, we present and discuss similar experiments using RPL [31] as a baseline. For more details, please refer to Appendix Table 7.

**Ablation Study of our CG-Aug**    The proposed CG-Aug generate semantic-shift and domain-shift jointly in a coherent way. To evaluate the design, we compare with three variations: (1) *Semantic-Shift Only (SS)*: Generate images with semantic shift using POC [12]. (2) *Domain-shift or Semantic-shift (DS or SS)*: Create a mixed dataset with either domain shifts (DS) using our semantic-mask-to-image process or semantic shifts (SS) using POC. (3) *Domain-shift and Semantic-shift (DS and SS)*: First generate DS data, then inpaint unknown objects. The second and third methods can be seen as applying [33] to our problem in two ways. Results in Table 4 show that: adding domain shift data

Table 4: **Ablation Study of CG-Aug**. Generating data with both Semantic-shift (SS) and Domain-shift (DS) in a coherent manner achieves better results than other variations. The experiments were conducted using the Mask2Former backbone and evaluated on the RoadAnomaly dataset.

| | AUC↑ | AP↑ | FPR$_{95}$↓ |
|---|---|---|---|
| POC [12] (SS) | 95.43 | 83.66 | 10.33 |
| DS or SS | 95.90 | 87.64 | 9.28 |
| DS and SS | 96.47 | 89.08 | 8.16 |
| CG-Aug (Ours) | **97.94** | **90.17** | **7.54** |

significantly improves performance over semantic-shift-only data. Jointly generating DS and SS in one image yields better results than generating them separately. Our method, which generates both DS and SS in one step, achieves the best performance, ensuring more coherence without artifacts and outperforming the two-step approach. We include more comparison results with POC [12] in Appendix C.3.

Table 5: **Abaltion Study of Our Training Pipeline**: Learnable Uncertainty Function (Learnable-UF), Relative Contrastive Loss (RelConLoss), and Noise-aware Sample Selection (Selection). Experiments are conducted under DeepLabv3+ architecture.

| Learnable-UF | RelConLoss | Selection | SMIYC-RA Val | | SMIYC-RO Val | | MUAD | | | | ACDC -POC | | | |
|---|---|---|---|---|---|---|---|---|---|---|---|---|---|---|
| | | | AP↑ | FPR$_{95}$↓ | AP↑ | FPR$_{95}$↓ | AP↑ | FPR$_{95}$↓ | mIoU↑ | mAcc↑ | AP↑ | FPR$_{95}$↓ | mIoU↑ | mAcc↑ |
| ✗ | ✓ | ✓ | 89.34 | 7.51 | 94.96 | 0.21 | 24.10 | 25.73 | 31.67 | 72.16 | 77.92 | 2.00 | 53.43 | 84.46 |
| ✓ | ✗ | ✓ | 91.01 | 5.78 | 94.95 | 0.20 | 20.49 | 24.58 | 32.68 | **73.86** | 75.67 | 1.60 | 53.22 | 84.32 |
| ✓ | ✓ | ✗ | 91.64 | 4.18 | **96.07** | **0.15** | 20.24 | 22.57 | 31.73 | 71.88 | 77.99 | 1.27 | 54.26 | **85.30** |
| ✓ | ✓ | ✓ | **93.82** | **3.94** | 95.20 | 0.19 | **36.08** | **18.74** | 31.33 | 73.13 | **82.41** | **1.01** | 54.12 | 85.07 |

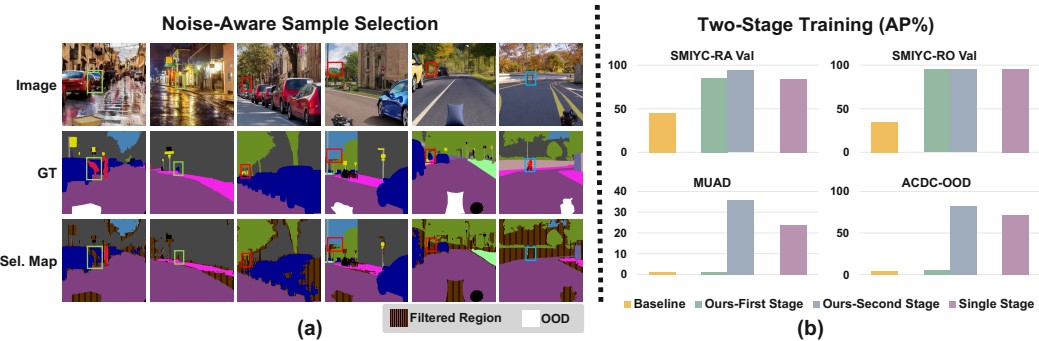

Figure 4: (a) **Visualization of Our Selection Maps.** Our selection strategy effectively identifies and removes generation errors (highlighted with boxes). (b) **Analysis of Our Two-Stage Training.** The first stage of training the uncertainty function boosts baseline performance, and second-stage fine-tuning further improves performance, achieving better results than single-stage training.

**Ablation study of our Training Design** We evaluate our training design in Table 5. In Row #1, we replace our *Learnable Uncertainty Function* (Learnable-UF) with a fixed energy function. In Row #2, we substitute our *Relative Contrastive Loss* (RelConLoss) with an absolute contrastive loss [46, 31, 42], which directly supervises the uncertainty score value rather than the relative gap between two uncertainty scores. In Row #3, we remove the *Sample Selection* module. Compared to our complete method, presented in the final row, these modifications result in decreased performance in both OOD detection and domain generalization, highlighting the effectiveness of our module design. A visualization of the sample selection process is shown in Figure 4 (a).

In Figure 4(b), we evaluate the effectiveness of our *Stage-wise Training* pipeline. Starting from a pre-trained baseline model, our first stage—fine-tuning only the learnable uncertainty function—doubles the performance on SMIYC-RA/RO datasets, demonstrating that the initial uncertainty function is often sub-optimal and can be significantly improved using fixed features [24]. A second-stage feature fine-tuning further boosts performance. Additionally, our two-stage approach outperforms single-stage fine-tuning with a learnable uncertainty function, showing that training directly with uncalibrated uncertainties can disrupt feature learning and degrade OOD detection performance.

We also demonstrate the robustness of our method under a range of hyperparameters (loss margins and selection ratio) in Appendix B.1, and evaluate the impact of generated dataset size in Appendix B.2.

## 5 Conclusion

In this work, we have studied semantic segmentation under multiple distribution shifts, finding that prior methods focusing separately on domain generalization and anomaly segmentation may not effectively handle these complex shifts. To tackle this, we have introduced a coherent generative data augmentation approach that enriches training data with both domain and semantic shifts. Additionally, we have proposed a learnable uncertainty function, trained in a stage-wise manner, to fully utilize the data and produce uncertainty scores specifically for semantic shifts. One limitation of our method is its reliance on the quality of the generative model. While we mitigate generation failures through offline autofiltering and online sample selection, some impact remains, such as lower performance for classes the generative model struggles with and potential limitations in scaling up the generated data (see Appendix E for details).

## Acknowledgement

This work was supported by NSFC 62350610269, Shanghai Frontiers Science Center of Human-centered Artificial Intelligence, and MoE Key Lab of Intelligent Perception and Human-Machine Collaboration (ShanghaiTech University).

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

# A  Implementation Details

## A.1  Zero-Shot Semantic-to-Image Generation

We adopt a pretrained semantic-to-image generation model, ControlNet 1.0 [53], provided by the official GitHub repository[4], for our generation process. This model is based on Stable Diffusion [43] and fine-tuned on ADE20K [54, 55]. It takes two main inputs: a semantic mask and a text prompt.

To obtain our pasted semantic mask, we first convert the masks from Cityscapes labels to ADE20K labels, then overlay this with auxiliary out-of-distribution (OOD) object masks. Specifically, we use the mask labels from ADE20K that belong to the 'thing' categories, excluding those with labels shared with Cityscapes.

Our text prompts have two parts: one part specifies the domain shifts, and the other specifies the OOD objects. For domain-shift prompts, we use the template "`An image sampled from various stereo video sequences taken by dash cam in {PLACE} in a {WEATHER} {TIME}`", where we define `PLACE` as a set of 100 cities worldwide, `WEATHER` as [`'cloudy'`, `'rainy'`, `'snowy'`, `'foggy'`, `'clear'`], and `TIME` as [`'day'`, `'night'`]. Additionally, we improve the OOD object generation by indicating the specific class of pasted objects in the prompt with the template: "`There is a {OOD} accidentally staying on the road.`", where `OOD` is the class name of the pasted object. This further contextualizes the generated scene to reflect realistic anomaly scenarios.

## A.2  Auto-Filtering of Failed Generations

The generation process can be noisy, particularly when generating image regions for pasted OOD object masks. By nature, these objects are anomalies within the scenes, appearing rarely, and their cut masks may exhibit shapes or poses inconsistent with their surroundings. As a result, the generated objects may deviate significantly in shape from the intended mask, be overlooked (blending into the surroundings), or be incorrectly generated as more common objects within the scene. Such discrepancies make the raw augmented image-label pairs too noisy for direct training.

To address these issues, we design an automatic filtering process to identify generation failures, such as cases where no object is generated or a known-category object is incorrectly produced. If the generated object is present and does not belong to a known category, we retain the image while revising the corresponding mask for the generated novel object. Otherwise, we discard the image and regenerate it. To implement this, we use the Segment-Anything Model (SAM), providing the bounding box location of the pasted mask as input to obtain a segmentation. We then compare the SAM output with the original mask, identifying it as a failure case if the Intersection over Union (IoU) is very low (below 0.7). Additionally, we employ a pre-trained segmentation model to produce an uncertainty score for the generated objects, filtering out those with very low uncertainty scores, as these are likely to have been misgenerated into a known-category object. This comprehensive filtering process effectively enhances the quality of the training dataset, making it better suited for effective model training.

## A.3  Training Details on Mask2Former Backbone

Following Mask2Anomaly [42], we train Mask2Former [8] using a combination of dice loss and binary cross-entropy (BCE) loss for the mask prediction head, and cross-entropy loss for the class prediction head. For the dice and BCE losses, we modify the sampling strategy for generated images to implement the sample selection process described in Sec. 3.3. Specifically, we compute the pixel-level BCE loss and select pixels with lower BCE losses for backpropagation in both the dice and BCE loss calculations. Since most generation errors occur at the pixel level, we do not apply sample selection for the mask-wise class prediction in the mask prediction head.

We maintain the same model architecture as Mask2Anomaly, which includes a ResNet-50 [19] backbone, a pixel decoder, a Transformer [47] decoder, and a global mask attention mechanism that independently distributes attention between foreground and background. As in Mask2Anomaly, we keep the ResNet backbone frozen while training the remaining model components, and we employ the

---

[4]`https://github.com/lllyasviel/ControlNet/tree/main`

AdamW [34] optimizer with its default learning rate and scheduler. For our method, the uncertainty loss margins are set to $\lambda_1 = 0.7$, $\lambda_2 = 0.5$, and $\lambda_3 = 0.2$, and we use a selection ratio $\alpha = 0.8$. The loss weights $\beta_1$ and $\beta_2$ are set to 10 for both. We use a batch size of 8 for all experiments and train the model on a single NVIDIA A40 48GB GPU.

### A.4 Training Details on DeepLabv3+ Backbone

For the DeepLabv3+ backbone, we follow the setup in RPL [31], using DeepLabv3+ with WideRes-Net38 pretrained by Nvidia. The backbone remains fixed, and only the ASPP layers are fine-tuned. We use the Adam [25] optimizer with a learning rate of 1.0e-6. For our method, the contrastive margins $\lambda_1, \lambda_2, \lambda_3$ are set to 10,5,5, the selection ratio is 0.8. The loss weights are 50 and 10 for $\beta_1$ and $\beta_2$ respectively. The batch size is set to 8, and all experiments are conducted on two NVIDIA A40 48GB GPUs.

### A.5 Rule-Based Augmentation

As a typical approach to domain generalization, we implement Rule-Based Augmentation as outlined in [45], using a set of image transformations. Specifically, we apply the following transformations, with their application probabilities indicated in parentheses: color jittering (0.5), Gaussian blur (0.5), random sharpness adjustment (0.5), random contrast adjustment (0.5), random equalization (0.5), random resizing (0.5), random rotation (0.5), random horizontal flipping (0.75), and random cropping (1.0).

## B  Additional Ablation Studies

### B.1  Impact of Hyperparameters

**Loss Margins**  Our relative contrastive loss 4 includes three terms, each with a margin value $\lambda$ controlling the distance penalty limits. These margins are set based on the average uncertainty scores from the training set. Specifically, we compute the differences in uncertainty scores between unknown vs. original known data, unknown vs. augmented known data, original known vs. augmented known data, and set the differences as margins for these distance respectively. Moreover, our two-stage training framework first trains the uncertainty function based on the existing model, allowing this function to adapt to different scales. This provides flexibility in parameter setting even without prior knowledge.

In Table 6, we evaluate the model's robustness across a wide range of hyperparameter variations. Our default loss margins are $[\lambda_1, \lambda_2, \lambda_3] = [10, 5, 5]$. We start by scaling them by 0.1 and 10 and conduct experiments using (1, 0.5, 0.5) and (100, 50, 50), respectively. As shown in Table 6(a), the uncertainty function adjusted to these scales with minimal impact on the results. Further analysis, such as changing the second and third parameters individually, showed that while the relative sizes of the three contrastive losses have some impact, the effect remains minor(see Table 6(b) and Table 6(c)). Ensuring that the parameters are set within an order of magnitude does not affect the results much.

Table 6: **Impact of Loss Margins.** We examine model robustness across various loss margins by evaluating margin scale impacts in (a) and analyzing effects of individual margins in (b) and (c). Results are reported on SMIYC-RA Val(AP & FPR) and MUAD (mIoU) using the DeepLabv3+ architecture.

<table>
<tr><td colspan="4">(a) Impact of margin scales.</td><td colspan="4">(b) Impact of margin $\lambda_2$.</td><td colspan="4">(c) Impact of margin $\lambda_3$.</td></tr>
<tr><td>Margins</td><td>Scale</td><td>AP↑</td><td>FPR↓  mIoU↑</td><td>$\lambda_2$</td><td>AP↑</td><td>FPR↓</td><td>mIoU↑</td><td>$\lambda_3$</td><td>AP↑</td><td>FPR↓</td><td>mIoU↑</td></tr>
<tr><td>[1,0.5,0.5]</td><td>x 0.1</td><td>93.81</td><td>5.16  **32.50**</td><td>3</td><td>92.34</td><td>**3.27**</td><td>**31.75**</td><td>3</td><td>92.53</td><td>**3.27**</td><td>**32.24**</td></tr>
<tr><td>[10, 5, 5]</td><td>x 1</td><td>93.82</td><td>3.94  31.33</td><td>5</td><td>**93.82**</td><td>3.94</td><td>31.33</td><td>5</td><td>**93.82**</td><td>3.94</td><td>31.33</td></tr>
<tr><td>[100,50,50]</td><td>x 10</td><td>**95.73**</td><td>**2.17**  29.61</td><td>7</td><td>91.52</td><td>4.27</td><td>31.28</td><td>7</td><td>92.45</td><td>**3.21**</td><td>31.73</td></tr>
<tr><td></td><td></td><td></td><td></td><td>10</td><td>91.75</td><td>4.85</td><td>30.91</td><td>10</td><td>92.08</td><td>3.96</td><td>31.56</td></tr>
</table>

**Selection Ratio** We determine the selection ratio for our sample selection process by visualizing the selection map of a small batch of data under several choices to ensure that visibly incorrect patterns are removed. To examine the impact of selection ratio to our method, we conducted experiments with selection ratios ranging from 0.6 to 0.9, as detailed in Figure 5 (a). The results show that while including too many pixels (1.0) introduces noise, and including too few (0.6) removes useful regions, the model performance is stable within a wide range (0.7 to 0.9), demonstrating the robustness of the model to this hyperparameter.

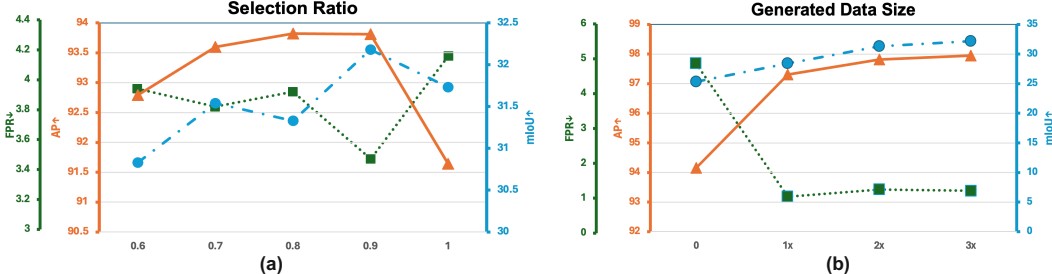

Figure 5: (a) **Impact of Sample Selection Ratio.** We report both anomaly segmentation performance(AP↑, FPR↓ on SMIYC-RA Val) and known class segmentation performance (mIoU↑ on MUAD). Experiments are conducted under DeepLab v3+ architecture. (b) **Impact of Generated Data Size.** We observe an improvement of performance with the increase of generated data size with the same evaluation under Mask2Former architecture.

## B.2 Impact of the Size of Generated Dataset

By default, we generate our distribution-shift dataset at the same size as Cityscapes. To analyze the impact of the generated dataset size, we scale it to 2x and 3x the size of the Cityscapes training set. As shown in Fig. 5(b), there is a significant improvement from dataset sizes 0 to 1, demonstrating the effectiveness of our generated data, with further gains observed as the dataset size increases.

## B.3 Impact of CG-Aug and Training Strategy for RPL

We evaluate the decoupled contributions of our data augmentation and training strategies with RPL [31] in Table 7. Similar to Table 3, we first replace its original OOD data, which utilizes cut-and-pasted COCO images, with our proposed CG-Aug in Row#2. However, we find the improvement is not as significant. This may be due to certain aspects of RPL's loss and training design being less suitable for our scenario. Firstly, RPL relies on the original network's predictions to supervise a learnable residual part. Since the original network does not generalize well to data with domain-shift, this results in imprecise supervision. Secondly, the RPL uncertainty loss focuses solely on increasing uncertainty for unknowns, without adequately addressing the known classes, particularly for augmented images. Additionally, restricting the trainable parameters to a residual block may limit the model's ability to learn more complex patterns, thereby reducing overall effectiveness.

Next, we replace their training strategy with ours, leading to consistent performance improvement. Those results demonstrate that effectively utilizing the generated training data with multiple distribution shifts remains an open question. Our work takes a step towards analyzing the shortcomings of existing training designs, offering novel and effective strategies for better handling this data.

# C  Additional Quantitative Results

## C.1 Additional Results on ACDC-POC

**Performance under Individual Domain Shifts** In addition to the main Table 2, we present the ACDC-POC results with domain-specific splits to provide a more detailed analysis of our method across different types of domain shifts. As shown in Table 8, our method outperforms previous state-of-the-art methods (RPL [31] and Mask2Anomaly [42]) across four domains—fog, rain, snow, and night—on most metrics in both OOD detection and known-class segmentation.

Table 7: **Impact of CG-Aug and Training Strategy.** We evaluate our proposed coherent generative-based augmentation on the previous OOD method, RPL [31], the improvement is not obvious. However, with our training strategy, the performance has largely improved. This demonstrates our training method can effectively utilize the generated training data with multiple distributions.

| | | RoadAnomaly | | SMIYC-RA Val | | SMIYC-RO Val | |
|---|---|---|---|---|---|---|---|
| Training | Aug. | AP↑ | FPR$_{95}$ ↓ | AP↑ | FPR$_{95}$ ↓ | AP↑ | FPR$_{95}$ ↓ |
| RPL [31] | Default | 71.61 | 17.74 | 88.55 | 7.18 | **96.91** | **0.09** |
| RPL [31] | Ours | 72.46 | 21.85 | 83.50 | 23.88 | 93.30 | 0.51 |
| Ours | Ours | **74.60** | **16.08** | **93.82** | **3.94** | 95.20 | 0.19 |

Table 8: **Results on ACDC-POC Datasets with specific domain-shift types**: Fog, Rain, Snow and Night. Our model achieves the best performance in both anomaly segmentation (AP↑, FPR↓ ) and domain-generalized segmentation (mIoU↑, mAcc↑ ).

| | Fog | | | | Rain | | | | Snow | | | | Night | | | |
|---|---|---|---|---|---|---|---|---|---|---|---|---|---|---|---|---|
| Method | AP↑ | FPR$_{95}$↓ | mIoU↑ | mAcc↑ | AP↑ | FPR$_{95}$↓ | mIoU↑ | mAcc↑ | AP↑ | FPR$_{95}$↓ | mIoU↑ | mAcc↑ | AP↑ | FPR$_{95}$↓ | mIoU↑ | mAcc↑ |
| RPL [31] | 88.3 | **0.6** | **71.2** | 93.4 | 67.4 | 1.9 | 50.9 | 88.0 | 75.1 | 2.5 | 49.0 | 84.5 | 74.3 | **0.8** | 24.4 | 51.2 |
| Ours (DeepLabv3+) | **89.7** | 0.7 | 69.1 | **94.3** | **73.6** | **1.7** | **59.9** | **92.5** | **77.9** | 2.0 | **55.4** | **89.3** | **83.7** | **0.8** | **35.4** | **65.3** |
| M2A [42] | 83.9 | **0.9** | 67.3 | **94.3** | 75.9 | 1.7 | 53.2 | 91.0 | 71.0 | 2.3 | 45.2 | 86.4 | 75.8 | 3.9 | 29.5 | 61.8 |
| Ours (Mask2Former) | **90.5** | **0.9** | **69.7** | 94.2 | **91.3** | **0.3** | **54.5** | **91.2** | **90.7** | **0.6** | **51.5** | **86.7** | **88.7** | **0.4** | **31.8** | 61.6 |

**Per-Class Segmentation Results** We evaluated per-class segmentation results and compared them with the baseline DeepLabv3+ [7] model. Results are presented in Table 9. Our method improves segmentation performance (mIoU) across most categories. However, performance in fence, pole, and traffic sign remains similar (with differences of less than 1%), and performance on vegetation decreases by 3%, likely due to poor generation quality for this class.

Table 9: **Per-class segmentation results.** We present the segmentation performance (mIoU) for each known class on the ACDC-POC dataset. Compared to the baseline model (DeepLabv3+ [7]), our method improves performance in most categories.

| Method | road | sidewalk | building | wall | fence | pole | traffic light | traffic sign |
|---|---|---|---|---|---|---|---|---|
| DeepLabv3+ [7] | 77.57 | 37.7 | 63.55 | 17.46 | **31.22** | 49.42 | 64.24 | 52.78 |
| Ours | **85.75** | **58.16** | **74.8** | **40.79** | 29.43 | **50.81** | **71.69** | **53.01** |

| | vegetation | terrain | sky | person | rider | car | truck | bus |
|---|---|---|---|---|---|---|---|---|
| DeepLabv3+ [7] | **75.75** | 10.58 | 81.12 | 54.17 | 19.37 | 77.89 | **50.76** | **50.74** |
| Ours | 72.82 | **30.21** | **81.35** | **62.77** | **32.17** | **79.99** | 49.41 | 49.33 |

| | train | motorcycle | bicycle |
|---|---|---|---|
| DeepLabv3+ [7] | 30.33 | 13.77 | 22.05 |
| Ours | **38.08** | **29.19** | **38.52** |

## C.2  Comprehensive Metric Results on SMIYC

To provide a comprehensive analysis of our method, we complement the main metrics (AP, FPR) with component-wise metrics (sIoU, PPV, F1) on the SMIYC benchmark. The results are summarized in Table 10. As shown, our method outperforms RPL [31] and Mask2Anomaly [42] across most evaluation metrics.

## C.3  Comparison with POC [12]

We present additional comparison results of our CG-Aug method against POC [12] across six datasets. Following the experimental setup in POC [12], we replace the default OOD data (COCO) in Mask2Anomaly [42] with our CG-Aug. As shown in Table 11, our method outperforms both POC variations on most datasets. Notably, for FS-Static [4], where OOD objects are introduced through

Table 10: **Comprehensive Metric Results on SMIYC.** We present the results of our method across pixel-wise metrics (AP, FPR) and component-wise metrics (sIoU, PPV, and F1). Compared to recent methods RPL [31] and Mask2Anomaly [42], our method achieves superior or comparable results.

| Method | Backbone | SMIYC-RA | | | | | SMIYC-RO | | | | |
|---|---|---|---|---|---|---|---|---|---|---|---|
| | | AP↑ | FPR$_{95}$ ↓ | sIoU↑ | PPV↑ | F1↑ | AP↑ | FPR$_{95}$ ↓ | sIoU↑ | PPV↑ | F1↑ |
| RPL [31] | DeepLab v3+ | 83.49 | 11.68 | 49.76 | 29.96 | 30.16 | 85.93 | 0.58 | **52.61** | 56.65 | 56.69 |
| Ours | | **88.06** | **8.21** | **56.15** | **34.66** | **37.83** | **90.71** | **0.26** | 48.13 | **66.73** | **58.02** |
| M2A [42] | Mask2Former | 88.70 | 14.60 | **60.40** | 45.70 | 48.60 | 93.30 | 0.20 | **61.40** | 70.30 | **69.80** |
| Ours | | **91.92** | **7.94** | 58.74 | **45.77** | **48.74** | **95.29** | **0.07** | 59.43 | **73.51** | 68.70 |

cut-and-paste, COCO achieves the best performance; however, this setup lacks the ability to reflect realistic OOD objects encountered in real-world scenarios.

Table 11: **Comparison of our CG-Aug with POC.** Following the experimental setup in POC [12], we replace the default OOD data (COCO) in Mask2Anomaly [42] with our CG-Aug, and compare with two versions of POC. Our method outperforms both POC variations on most datasets.

| Mask2Anomaly+ | Road Anomaly | | SMIYC-RA21 (val) | | SMIYC-RO21 (val) | | ACDC-POC | | FS-L&F (val) | | FS-Static (val) | |
|---|---|---|---|---|---|---|---|---|---|---|---|---|
| | AP↑ | FPR↓ | AP↑ | FPR↓ | AP↑ | FPR↓ | AP↑ | FPR↓ | AP↑ | FPR↓ | AP↑ | FPR↓ |
| COCO | 79.70 | **13.45** | 94.50 | 3.30 | 88.6 | 0.30 | 73.77 | 3.60 | 69.41 | 9.46 | **90.54** | **1.98** |
| POC-c [12] | 82.3 | 36.7 | 93.8 | 2.1 | 95.3 | 0.3 | 74.5 | 7.6 | 68.8 | 11.4 | 87.4 | 3.1 |
| POC-alt [12] | 78.0 | 24.6 | 92.1 | 8.4 | **96.0** | **0.1** | 72.0 | 8.4 | 73.0 | **9.2** | 87.0 | 2.1 |
| CG-Aug (Ours) | **85.47** | 22.38 | **97.96** | **1.55** | 89.80 | 0.12 | **86.17** | **1.05** | **76.56** | 10.17 | 85.70 | 7.16 |

## C.4 Comparison with DG Methods on the Original ACDC Dataset

To assess the effectiveness of our method in domain generalization, we conducted additional comparisons with several recent approaches, including IBN [39], IterNorm [22], IW [40], ISW [10], ISSA [27], and CMFormer [3], on the ACDC [44] dataset. As shown in Table 12, our method outperforms all ResNet-based methods in the Fog, Rain, and Snow domains, achieving comparable results in the Night domain. Among Mask2Former-based methods, our approach also surpasses ISSA [27], which similarly uses a ResNet backbone. However, there remains a notable performance gap between our method and CMFormer [3], likely due to CMFormer's use of the Swin Transformer [32] as the backbone for Mask2Former.

Table 12: **Domain generalization performance comparison between our method and other DG methods.** Results from other methods are taken from CMFormer [3]. All methods are trained on the Cityscapes [11] dataset and tested on the ACDC [44] dataset. Results are shown in mIoU (%).

| Method | Backbone | Fog | Night | Rain | Snow | Mean |
|---|---|---|---|---|---|---|
| IBN[39] | | 63.8 | 21.2 | 50.4 | 49.6 | 43.7 |
| Iternorm[22] | | 63.3 | 23.8 | 50.1 | 49.9 | 45.3 |
| IW[40] | ResNet | 62.4 | 21.8 | 52.4 | 47.6 | 46.6 |
| ISW[10] | | 67.5 | **33.2** | 55.9 | 53.2 | 52.5 |
| Ours | | **74.84** | 30.33 | **61.15** | **57.80** | **55.95** |
| ISSA[27] | | 67.5 | 33.2 | 55.9 | 53.2 | 52.5 |
| CMFormer[3] | Mask2Former | **77.8** | **33.7** | **67.6** | **64.3** | **60.1** |
| Ours | | 70.85 | 29.39 | 54.54 | 52.97 | 51.61 |

## D Visualization of Generated Data

In Fig. 6, we provide additional visualization examples of our generated images (row 2), corresponding selection maps (row 3), and the loss maps used to produce the selection map (row 4). Below each column, we display the weather, time, and location prompts that guide the model in generating

diverse covariate shifts, as well as the OOD prompts used to generate objects. As shown, our method effectively generates images with both domain and semantic shifts, with the novel objects blending seamlessly into the background (e.g. pose and lighting). Additionally, our sample selection process effectively filters out some generation errors (highlighted in red boxes).

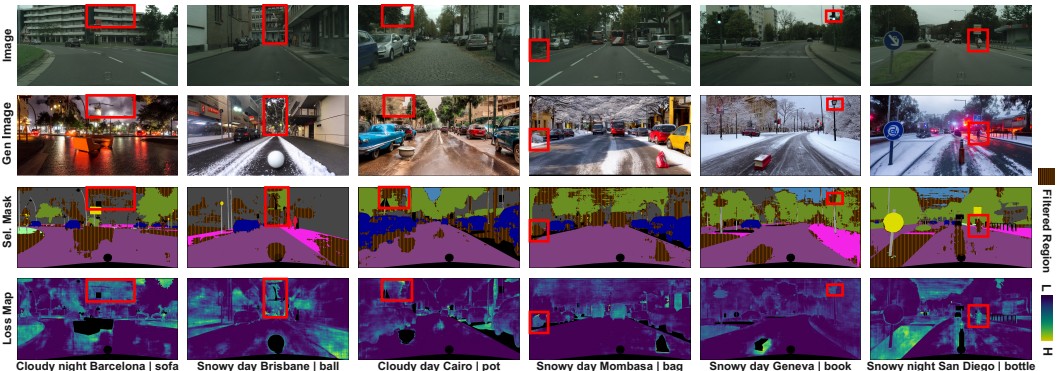

Figure 6: **Visualization of Generated Images.** Row 1: Original images from Cityscapes. Row 2: Generated images featuring both semantic and domain shifts. Row 3: Selection map used to calculate selected cross-entropy loss during training. Row 4: Cross-entropy loss map used to produce the selection map (excluding the OOD regions, which are not involved in known class segmentation loss calculation). Below each column, we display the weather, time, and location prompts that guide the model in generating diverse covariate shifts, along with the OOD prompts for object generation. Red boxes highlight generation errors.

# E    Discussion of Generation Failures and Their Impact

A limitation of our method is its reliance on the quality of the generative model. Although we apply offline auto-filtering and online sample selection to minimize the impact of generation failures during training, some issues may still arise. Specifically, we observe that generation failures typically occur in the following scenarios: (a) remote scenes, (b) small objects, and (c) text-related elements. These limitations highlight the current constraints of generative models and suggest areas for future research. Below, we discuss the impact of generation failures:

**Impact on Class-Specific Learning:**    Generation failures can adversely affect specific classes. As shown in Table 9, we evaluated per-class segmentation results and compared them with the baseline model on Cityscapes. Performance in categories—such as fence, pole, and traffic sign—remains similar (differences of less than 1%), and vegetation shows a 3% decrease, likely due to lower generation quality for these classes.

**Performance Saturation:**    We observe that performance tends to saturate with increasing amounts of generated data. Experiments with dataset scaling from 1.0x to 2.0x and 3.0x Cityscapes sizes, as shown in Figure 5, indicate that while performance improves with larger dataset size, it eventually plateaus. This saturation may result from an interplay between the benefits of additional data and the adverse effects of generation failures.

# F    Societal Impacts

Enhancing OOD detection in autonomous vehicles can significantly improve safety by enabling these systems to better recognize and respond to novel and unexpected situations, thereby reducing the risk of accidents. Improved robustness to domain shifts also contributes to greater resilience and safety across diverse driving scenarios. However, improved OOD detection may lead to an over-reliance on autonomous systems, potentially reducing the vigilance of human drivers or passengers in semi-autonomous vehicles. Additionally, unintended biases in OOD detection systems could result in unsafe responses to certain situations, particularly if training data does not sufficiently cover diverse scenarios, potentially compromising safety in rare but critical cases.

