# OpenReview forum: "Generalize or Detect? Towards Robust Semantic Segmentation Under Multiple Distribution Shifts"
_NeurIPS.cc/2024/Conference — NeurIPS 2024 poster_

### Official Review · Reviewer_ua5t · 2024-06-13

**Soundness:** 2
**Presentation:** 2
**Contribution:** 3
**Rating:** 6
**Confidence:** 4

**Summary:**

The authors propose a method which generalize effectively to covariate-shift regions while precisely identifying semantic-shift regions, i.e., domain generalization and OoD segmentation. They design a novel generative augmentation method to produce coherent images that incorporate both, various covariate shifts and anomaly objects. Moreover, they introduce a training strategy that recalibrates uncertainty specifically for semantic shifts and enhances the feature extractor to align features associated with domain shifts. The approach is compared  across different benchmarks for domain generalization and OoD segmentation (or both).

**Strengths:**

The paper is written clearly. The authors address an important problem, namely domain genralization and the detection of unknown objects in one step. While most works focus on one of these robustness problems, the authors present a method that tackles both together. The approach of using the power of generative models to obtain further (augmented) training data to increase the robustness of the network is good. Furthermore, calibrating uncertainty, i.e., generating high OoD scores for semantic-shift regions and performing robustly under covariate shifts, is a sound approach. The method outperforms the given baselines.

**Weaknesses:**

The paper is written in a clear way, but it is also a bit unclean, for example the caption of figure 1c, the reference to figure 2 is missing in the text and there are a few typos (for example "the ResNet backbone are froze" or "AadmW").

The related work section is rather short and as a comparison to anomaly detection and domain generalization it is only mentioned that these methods only focus on their problem, but not on both (in comparison to the authors). However, I miss a methodological comparison at this point to highlight the novelties of the paper.

In the SMIYC benchmark, there are other methods that perform better than the method presented, but there is no comparison with them or an argument of why a comparison does not make sense. In addition, 5 different metrics are used in this benchmark for the evaluation, but the authors only calculate 1, which makes the comparison more difficult.

The method depends on many hyperparameters, but there are no ablation studies on different values.

**Questions:**

The described papers for domain generalization are very limited and rather older, there is certainly newer literature to compare with.

The problem of domain generalization is considered, but the SMIYC validation data is used for model selection, so is it domain adaptation?

**Limitations:**

No limitations section in the paper, but the robustness of the method was demonstrated on various datasets.

---

> ### Author Response · Authors · 2024-08-07
> **Rebuttal by Authors**
>
> We thank the reviewer for the time and detailed feedback. We address the SMIYC benchmark comparison in the general response and will revise the paper to correct the noted typos. Below, we address your specific concerns.
>
> # Weakness 2: Related Work & Novelty
> **Anomaly Segmentation:** Our method falls under discriminative-based anomaly segmentation methods that utilize additional out-of-distribution (OOD) data to train models to distinguish between known and unknown data [3, 30, 21, 26]. We contribute to the construction of generative-based OOD data and training pipelines. The details of our OOD data generation are explained in the general response. Regarding training, previous works [3, 30, 26] directly train models using OOD score functions calculated on logits (e.g., MSP, Energy, Entropy). In contrast, our approach introduces a learnable uncertainty function, initially set as the standard OOD score function and optimized before fine-tuning the feature extractor for joint known class segmentation and OOD detection. This approach decouples the training process to mitigate competition and leverages the meaningful features learned from previous closed-world pretraining.
>
> **Domain Generalization:** Existing methods for domain generalization include instance normalization or whitening [24, 6, 25] and domain-invariant feature learning through domain randomization [33, 29]. Our contribution lies in the latter category, where we propose a generative-based data augmentation technique. Unlike recent works that use generative models for feature [d1] or image randomization [d2], our technique concurrently generates data with both domain shifts and unknown objects. We believe that incorporating various distribution shifts is crucial to avoiding model biases and better handling domain and semantic shifts.
>
> # Weakness 3: Evaluation Metrics
>
> We would like to clarify that we used two metrics, AP and FPR@95, in the presented SMIYC results. These metrics are widely recognized as the **primary evaluation criteria** in the anomaly segmentation literature [1]. It is a common practice in this field to present only the primary metrics in the main paper due to space constraints, as exemplified in [21, d3].
>
> We agree with the reviewer that using multiple metrics provides a more comprehensive evaluation of the method. To address the reviewer’s concern, we have included the results of our method across all five metrics below. As shown in Table A1, our method consistently performs better or on par with RPL/Mask2Anomaly across all evaluation metrics. These additional results will be included in the appendix of the revised paper.
>
> Table A1: Comparison of our methods with two recent works RPL and Mask2Anomaly on the SMIYC benchmark.
> | Exp Name | Backbone | SMIYC-RA21 |  |  |  |  | SMIYC - RO21 |  |  |  |  |
> | --- | --- | --- | --- | --- | --- | --- | --- | --- | --- | --- | --- |
> |  |  | AP$\uparrow$ | FPR95$\downarrow$ | sIoU$\uparrow$ | PPV$\uparrow$ | F1$\uparrow$ | AP$\uparrow$ | FPR95$\downarrow$ | sIoU$\uparrow$ | PPV$\uparrow$ | F1$\uparrow$ |
> | RPL (ICCV 23) | DeepLab v3+ | 83.49 | 11.68 | 49.76 | 29.96 | 30.16 | 85.93 | 0.58 | 52.61 | 56.65 | 56.69 |
> | Ours |  | 88.1 | 8.2 | 56.2 | 34.7 | 37.8 | 90.7 | 0.3 | 48.1 | 66.7 | 58.0 |
> | Mask2Anomaly (ICCV 23) | Mask2Former | 88.70 | 14.60 | 60.40 | 45.70 | 48.60 | 93.30 | 0.20 | 61.40 | 70.30 | 69.80 |
> | Ours |  | 91.9 | 7.9 | 58.7 | 45.8 | 48.7 | 95.3 | 0.1 | 59.4 | 73.5 | 68.7 |
>
> # Weakness 4: Hyperparameters
> We thank the reviewer for the suggestion. We have added results ablating the two types of hyperparameters in our method: selection ratio and loss margins. THe results shown in Table 1 of the attached PDF demonstrate the robustness of our model to various hyperparameters.
>
> [d1] Gong, Rui, et al. Prompting diffusion representations for cross-domain semantic segmentation. BMVC (2024).
> [d2] Jia, Yuru, et al. Domain-Generalizable Semantic Segmentation with Image Diffusion Models and Stylized Semantic Control. ECCV (2024).
> [d3] Grcic et al., On Advantages of Mask-level Recognition for Outlier-aware Segmentation, CVPRW 2023.

---

> ### Author Response · Authors · 2024-08-07
> **Rebuttal by Authors**
>
> # Question 1: Domain Generalization Papers
> We chose RobustNet (2021)[6] and RuleAug (2023)[29] for our benchmark comparison because they represent two main DG strategies: Constraining the learned feature distributions and domain randomization. Although adding more recent DG techniques could enhance our analysis, many of these works do not provide pretrained models on Cityscapes or are based on different segmentation backbones, making direct evaluation and comparison challenging [25, d5].
>
> To address the reviewer’s concern, we additionally evaluated a recently published work, CMFormer [d4] (2024), which adopts the Mask2Former architecture for DG in semantic segmentation. Using the provided pretrained model and official code, we performed inference on the ACDC-POC dataset and compared it with our method under the Mask2Former architecture. We note that the comparison is not entirely fair, as CMFormer uses a more powerful Swin Transformer backbone while we use ResNet50 following Mask2Anomaly. As shown in Table A2,  CMFormer non-surprisingly performs better on known class segmentation with a 10-point gap in mIoU and 2 points in mAcc. However,  its OOD segmentation performance is significantly worse, with over 60 points lower in AP and 30 points higher in FPR. This supports our finding that existing DG techniques might overly generalize to all types of distribution shifts, making it difficult to recognize unknown objects, raising safety concerns in autonomous driving scenarios.
>
> | Domain | ACDC-POC |  |  |  |
> | --- | --- | --- | --- | --- |
> | Method | AP$\uparrow$ | FPR$\downarrow$ | mIoU$\uparrow$ | mAcc$\uparrow$ |
> | CMFormer | 27.84 | 31.25 | **60.71** | **85.90** |
> | Ours | **90.42** | **0.46** | 51.75 | 83.16 |
>
> # Question 2: Validation
> The main difference between domain generalization and domain adaptation techniques is that the former focuses on training one model to generalize to any data with different domain shifts, while the latter typically trains each model for each target domain. Our method belongs to domain generalization as we train one model on Cityscapes and evaluate it on various datasets with different domain shifts. We use the SMIYC validation set for model selection to ensure a fair comparison with other anomaly segmentation techniques in the benchmark, which also use the validation set for model selection.
>
> [d4] Bi, Qi et al. Learning content-enhanced mask transformer for domain generalized urban-scene segmentation. AAAI, 2024.
> [d5] Li, Yumeng et al. Intra-Source Style Augmentation for Improved Domain Generalization. WACV, 2023.

---

> > ### Comment · Reviewer_ua5t · 2024-08-10
> >
> > Thank you for the additional experiments and the responses. Given the new experiments and answers regarding all reviewers, I have increased my score.

---

> > > ### Author Response · Authors · 2024-08-13
> > > **Thank you**
> > >
> > > Thank you for increasing the score and for your time and careful review. We are glad to have addressed your concerns with the additional experiments and discussions, and we will incorporate these into the final manuscript.

---

### Official Review · Reviewer_gYbq · 2024-07-01

**Soundness:** 3
**Presentation:** 3
**Contribution:** 3
**Rating:** 6
**Confidence:** 4

**Summary:**

This work proposes a novel generative pipeline and fine-tuning method for anomaly detection under domain shift. The generative pipeline uses a semantic-map to image model that can leverage the labels from the Cityscapes dataset with some modifications which introduce novel unknown classes. The resulting images have unknown novel objects (semantic shifts) and modified known classes (covariate shifts) while preserving most of the semantic meaning.

Authors use the images augmented with their pipeline to train with a contrastive loss that pushes the representations of known classes together while pushes the novel categories away.

**Strengths:**

Applying a generative pipeline to augment images with either covariate or semantic shifts while trying to be as realistic as possible is an interesting direction with potential impact in the anomaly segmentation community.

Moreover, how to best leverage the generated images explicitly during training is also an interesting avenue to explore.

Results presented show convincing improvements upon recent work.

**Weaknesses:**

Although experiments are quite extensive, it would have been interesting to decouple the contributions of the generative pipeline and the proposed fine-tuning mechanism in the experiments. Many of the methods in table 2 could have used the images augmented with the generative model, and it would be very interesting to see how much can the generated data benefit those models.

Especially because it seems from Table 2 that on the MUAD dataset, the OOD+RuleAug baseline is worse than RPL or Mask2Anomaly and to my understanding, the OOD+RuleAug baseline consists in the training proposed in section 3.3 but replacing the generative data augmentation with a rule-based one from [29]). Thus, leaves the question whether other existing methods like RPL or Mask2Anomaly + the proposed generated data could be better than the training scheme proposed in 3.3.

I am aware the authors show in table 3 that every proposed component in the training pipeline leads to a drop in accuracy when removed, however, that does not show that the overall proposed pipeline is better than that of previous works if they all used the generated images for training (I would only focus on comparing with RPL / Mask2Anomaly which are the most recent ones).

Moreover, comparing with previous methods, using a contrastive loss was already proposed in prior work e.g. RPL[21]. I think a better discussion of the related work in section 3.3 to highlight the novelty in the components would be very helpful to readers.

In terms of the generative pipeline, I would like to bring to the attention of the authors a preprint which is very much aligned with the proposed pipeline and that it might be worth discussing in the related work:

[Loiseau *et al.* Reliability in Semantic Segmentation: Can We Use Synthetic Data? ArXiv 2023](https://arxiv.org/pdf/2312.09231)

As a preprint authors are not expected to compare quantitatively but given the similarity and the fact it was published Dec 2023 might be fair to mention it in the related work and discuss the differences with [8] and [34] a bit more extensively.

Last but not least, although the proposed generative pipeline leverages a model that is conditioned on a semantic mask, the generative models still have important limitations and can introduce significant changes that do not align well with the original image. From looking at Fig 5 in the appendix, I could easily spot that in the first image it removes a very large building and replaces it mostly with sky; in the second it does something similar replacing the building at the end with trees; the fifth one replaces traffic lights with street lamps and in the last image modifies the direction sign with a "sign" that does not mean anything. Perhaps for some applications this will not matter, depending on the classes of interest and in the level of semantic detail one might want to accomplish, but in the context of autonomous this kind of modifications might be problematic.

I think the pipeline still has its merit and can be of use to the community but a more clear discussion of the limitations of generative models is needed and perhaps some examples of failure cases would be of interest to add in the appendix.

**Questions:**

See my weaknesses section. Especially I'd be interested to know what are the main differences of the proposed training schedule and RPL and Mask2Anomaly and the main differences between the generative pipeline and [8, 34] and [Loiseau *et al.*](https://arxiv.org/pdf/2312.09231).

**Limitations:**

See my last point in the weaknesses section. More discussion on the limitations of the generative pipeline, especially regarding the preservation of semantic details.

---

> ### Author Response · Authors · 2024-08-07
> **Rebuttal by Authors**
>
> Thank you for the thorough comments and many constructive suggestions. We appreciate the mention of the interesting work by Loiseau et al. [a1] and have discussed our differences, including [8] and [34], in the general response. We address the other concerns below.
>
> # 1. Decouple Contribution
> We thank the reviewer for the suggestions and have conducted additional experiments on Mask2Anomaly and RPL. The results are shown in Table C1. Below, we discuss the findings:
>
> -**Mask2Anomaly**: We observe consistent improvements across all datasets, demonstrating the benefits of using our generated data. However, the final performance achieved by Mask2Anomaly + CG-Aug is still lower than the model fine-tuned with our pipeline. This highlights the efficacy of the proposed training design.
>
> -**RPL**: The improvement is not as significant. This may be due to certain aspects of RPL's loss and training design being less suitable for our scenario. Firstly, RPL relies on the original network's predictions to supervise a learnable residual part. Since the original network does not generalize well to data with domain-shift, this results in imprecise supervision. Secondly, the RPL uncertainty loss focuses solely on increasing uncertainty for unknowns, without adequately addressing the known classes, particularly for augmented images. Additionally, restricting the trainable parameters to a residual block may limit the model's ability to learn more complex patterns, thereby reducing overall effectiveness.
>
> Those results demonstrate that effectively utilizing the generated training data with multiple distribution shifts remains an open question.  Our work takes a step towards analyzing the shortcomings of existing training designs, offering novel and effective strategies for better handling this data.
>
> Table C1: We apply our coherent generative-based augmentation (CG-Aug) to the most recent works, RPL and Mask2Anomaly.
>
> |  |  |  | RoadAnomaly |  | SMIYC-RA21(val) |  | SMIYC-RO21(val) |  |
> |---|---|---|---|---|---|---|---|---|
> | Backbone | FT | OOD Data | AP$\uparrow$  | FPR$\downarrow$ | AP$\uparrow$  | FPR$\downarrow$ | AP$\uparrow$  | FPR$\downarrow$ |
> | Mask2Former | Mask2Anomaly  | COCO (Default) | 79.70 | 13.45 | 94.50 | 3.30 | 88.6 | 0.30 |
> |  | Mask2Anomaly  | CG-Aug (Ours) | 85.47 | 22.38 | 97.96 | 1.55 | 89.80 | 0.12 |
> |  | Ours | CG-Aug (Ours) | **90.17** | **7.54** | **97.31** | **1.04** | **93.24** | **0.14** |
> | DeepLabv3+ | RPL  | COCO (Default) | 71.61 | 17.74 | 88.55 | 7.18 | **96.91** | **0.09** |
> |  | RPL  | CG-Aug (Ours) | 72.46 | 21.85 | 83.50 | 23.88 | 93.30 | 0.51 |
> |  | Ours | CG-Aug (Ours) | **74.60** | **16.08** | **93.82** | **3.94** | 95.20 | 0.19 |
>
>
> # 2. Contrastive Loss
>
> We appreciate the reviewer’s kind suggestion and will add more related work in our model training section (3.3). Below, we discuss the novelty of the proposed contrastive loss.
>
> **Compared with contrastive loss in RPL**: RPL employs their contrastive loss *on a projected feature space*, supervising OoD training with a combination of feature contrastive loss and an additional energy loss to maximize uncertainty scores for unknown class data. In contrast, we calculate our contrastive loss *directly on the uncertainty scores*. This direct supervision allows for more explicit and effective ranking of uncertainties across different data types. Experiments in Table C1 replacing our OOD loss with RPL's showed that RPL's feature contrastive loss alone barely improves performance. Even with combined losses, RPL's performance falls short of ours, demonstrating the efficacy of our direct supervision on uncertainty scores.
>
> **Compared with other OOD Losses**: Existing OOD losses either maximize uncertainty scores solely for unknown data or supervise unknown and known data separately. In contrast, our method supervises the relative distance between OOD and inlier samples. We find this contrastive term to be more robust to hyperparameters and easier to optimize. To demonstrate its efficacy, we replaced the distance-based supervision with value-based supervision, similar to that used in Mask2Anomaly and PEBAL. As shown in Table 3, row #3, this change led to a performance decrease, validating the effectiveness of our loss design.
>
> Table C2: Ablation study of our contrastive loss (In DeepLab v3+). Our loss performs better in both OOD detection and known class segmentation results.
> |  | SMIYC-RA21(val) |  | ACDC-POC |  |  |  |
> | --- | --- | --- | --- | --- | --- | --- |
> | Loss | AP $\uparrow$ | FPR$\downarrow$ | AP$\uparrow$ | FPR$\downarrow$ | mIoU$\uparrow$ | mAcc$\uparrow$ |
> | FeaConLoss  (RPL) | 68.40 | 48.42 | 30.98 | 40.20 | 49.42 | 79.38 |
> | FeaConLoss +Energy (RPL) | 86.41 | 9.29 | 76.12 | 1.93 | 51.43 | 83.15 |
> | RelConLoss (Ours) | **93.82** | **3.94** | **82.41** | **1.01** | **54.12** | **85.07** |

---

> ### Author Response · Authors · 2024-08-07
> **Rebuttal by Authors**
>
> # 4.Generation Failures
> We agree with the reviewer that generative models still have important limitations, and appreciate the reviewer’s detailed examination and careful thought for generation failure cases and impact. Below, we first discuss how we currently deal with generation failures, followed by our analysis of the generation failure pattern, and examine how these failures affect model training.
>
> - **Noise-aware Learning:** During training, we mitigate the impact of generation failures by using a sample selection strategy (Sec. 3.3.2). We note that for failure cases mentioned by the reviewer, some can be prevented via pixel selection during training. To better illustrate this, we include the loss and selection map corresponding to the images from Fig 5. in the attached PDF.
>
> - **Generation Failure Cases:** Inspired by the reviewer, we examine our generated image and observe that generation failures typically occur in the following scenarios: (a) Remote scenes. (b) Small objects. (c) Text-related elements. This demonstrates the limitations of the current generative model and we hope they can be addressed in future research.
>
> - **Impact of Generation Failures:**
>     - **Impact on Category Learning:** Generation failures may adversely affect specific classes. We evaluated per-class segmentation results and compared them with the baseline model. Results are presented in Table C3.  We find performance in six categories, such as fence, pole, and traffic sign, remains similar (differences less than 1%); and performance on vegetation is worsened by 3%, likely due to poor generation quality for this class.
>     - **Performance Saturation:** We observe that performance tends to saturate with increasing generative data. Experiments varying the dataset scale from 1.0x to 2.0x and 3.0x Cityscapes sizes, as shown in Figure 1(b) in the supplementary material, suggest this saturation. This may be due to an interplay between the benefits of additional data and the negative impact of generation failures
>
> We will include more examples of failure cases and a comprehensive discussion of the limitations of generative models in the revised paper and appendix.
>
> Table C4. Per-class segmentation results in mIoU on ACDC-POC dataset.
> |  | road | sidewalk | building | wall | fence | pole | traffic light | traffic sign | vegetation | terrain | sky | person | rider | car | truck | bus | train | motorcycle | bicycle |
> | --- | --- | --- | --- | --- | --- | --- | --- | --- | --- | --- | --- | --- | --- | --- | --- | --- | --- | --- | --- |
> | DeepLab | 77.57 | 37.70 | 63.55 | 17.46 | **31.22** | 49.42 | 64.24 | 52.78 | **75.75** | 10.58 | 81.12 | 54.17 | 19.37 | 77.89 | **50.76** | **50.74** | 30.33 | 13.77 | 22.05 |
> | Ours | **85.75** | **58.16** | **74.80** | **40.79** | 29.43 | **50.81** | **71.69** | **53.01** | 72.82 | **30.21** | **81.35** | **62.77** | **32.17** | **79.99** | 49.41 | 49.33 | **38.08** | **29.19** | **38.52** |

---

> > ### Comment · Reviewer_gYbq · 2024-08-09
> > **Reviewer response**
> >
> > I would like to thank the authors for their detailed response. After reading their response and other reviewers comments I think the authors did a reasonable job at addressing my questions and the other reviewers comments. Therefore I am inclined to keep my positive score.
> >
> > **Missing comparisons:**
> >
> >  I agree with reviewer LCzB that it would be nice to include RbA as it has a strong performance on SMIYC and is concurrent to Mask2Anomaly and RPL in ICCV 2023 so it might not be fair to exclude it from the comparison. It would also be nice to add Grcic et al., On Advantages of Mask-level Recognition for Outlier-aware Segmentation, against the comparisons. To me this has been sufficiently addressed with (the second) Table A1, which I think should be Table A2 :)
> >
> > **Novelty of paper:**
> >
> >  As I see it there are two main contributions in this work, one is the generative pipeline to augment images with both domain and semantic shifts and the other is the training pipeline to leverage that data. Initially the two were entangled as there were no experiments with other methods and their generated data. However, with the additional experiments in Table C1 it is more clear that, although other methods might benefit from the generated data in some cases, the proposed method seems to leverage the generated data better (i.e. Ours with CG-Aug (ours) is better than RPL/Mask2Anomaly with CG-Aug (ours) ).
> >
> > There are no experiments with the proposed fine-tuning method and COCO data as OOD which would have been interesting too to understand if the proposed fine-tuning method alone could surpass previous works or if the combination with the new generated data is the key to the improved performance.
> >
> > Regarding the comparison with POC [8], it would be interesting to discuss that the motivation in [8] is to precisely avoid domain shifts in the evaluation, as this would hinder anomaly segmentation. All datasets used in the comparison in Table 2 of the rebuttal pdf (RoadAnomaly, SMIYC and ACDC-POC) have strong domain shifts with respect to Cityscapes, but datasets with a smaller domain shift such as Lost and Found were not included. The proposed CG-Aug, combines both domain shifts and semantic shifts to improve anomaly segmentation out-of-domain while POC focuses on introducing semantic shifts that minimally modify the image beyond the introduced objects. Both will have its Pro's and Con's and this is where I see the novelty in the generative part.

---

> ### Author Response · Authors · 2024-08-13
> **Thank you and Further Explanation**
>
> Thank you for your constructive and detailed feedback.
>
> - **Regarding the comparisons**: We appreciate your suggestion and will include RbA and M2F-EAM (Grcic et al.) in the revised manuscript. Thank you also for pointing out our typo in the general response, where (the second) Table A1 should be Table A2.
> - **Regarding the novelty:** Thank you for acknowledging the contributions of our work. We are pleased that our additional experiments have clarified the advantage of our training pipeline, which is tailored to our specific task and effectively leverages our generated data.
>     - **Further experiments with COCO data as OOD**: In response to your further suggestion, we have conducted experiments using COCO data as OOD to assess the efficacy of our proposed training pipeline without the generated data. We note that some training components, such as noise-aware learning and relative contrastive loss, are tailored for scenarios involving the generated domain-shift data. Therefore, in this evaluation, we focus primarily on assessing the effectiveness of our two-stage learnable uncertainty function and the relative contrastive loss applied between the original data and OOD data. **The results in Table C5 show that our training method surpasses RPL and Mask2Anomaly on most metrics,** and the combination with our generative data yields even greater performance improvements.
>     - **Comparison with POC:** We appreciate your recognition of the novelty of our generative pipeline. Our work primarily addresses scenarios where both domain shifts and semantic shifts are present, which we believe are common in real-world applications. To address your concerns regarding performance under smaller domain shifts, we have included results on the FS LostAndFound and FS Static datasets, comparing our method with POC using the Mask2Anomaly training pipeline. As shown in Table C6, **our method demonstrates superior performance on FS LostAndFound**, indicating that our generated data closely resembles real OOD scenarios. However, on the FS Static dataset, our performance is lower than that of POC. We note that the FS Static dataset's OOD data is generated using a cut-and-paste technique, which may not reflect real-world OOD distributions. This could explain why simple COCO-pasted data achieves the best results on this dataset.
>
>     Overall, we thank you again for your valuable input, which has helped us clarify the components of our design. We will incorporate all these discussions into the revised paper.
>
>     Table C5: Performance of our training pipeline using COCO data, compared to previous methods Mask2Anomaly and RPL.
>
>     | Backbone | FT | OOD Data | Road Anomaly |  |  | SMIYC -RA21 (val) |  | SMIYC -RO21 (Val) |  |
>     | --- | --- | --- | --- | --- | --- | --- | --- | --- | --- |
>     |  |  |  | AUC$\uparrow$ | AP$\uparrow$ | FPR$\downarrow$ | AP$\uparrow$ | FPR$\downarrow$ | AP$\uparrow$ | FPR$\downarrow$ |
>     | Mask2Former | Mask2Anomaly | COCO | - | 79.70 | 13.45 | 94.50 | 3.30 | 88.60 | 0.30 |
>     |  | Ours | COCO | 95.83 | 80.94 | 29.12 | **97.41** | 1.60 | 92.89 | 0.50 |
>     |  | Ours | Ours | **97.94** | **90.17** | **7.54** | 97.31 | **1.04** | **93.24** | **0.14** |
>     | DeepLabv3+ | RPL | COCO | 95.72 | 71.61 | 17.74 | 88.55 | 7.18 | **96.91** | **0.09** |
>     |  | Ours | COCO | 96.26 | **76.01** | 17.44 | 91.40 | 6.80 | 96.77 | 0.12 |
>     |  | Ours | Ours | **96.40** | 74.60 | **16.08** | **93.82** | **3.94** | 95.20 | 0.19 |
>
>     Table C6: Comparison of our CG-Augmentation with POC on FS LostAndFound and FS Static, integrated into the Mask2Anomaly training pipeline. COCO results are from the official Mask2Anomaly paper;  POC alt. and POC c. results are from the official POC publication.
>
>     |  | FS_LostAndFound (val) |  | FS_Static (val) |  |
>     | --- | --- | --- | --- | --- |
>     | Mask2Anomlay+ | AP$\uparrow$ | FPR$\downarrow$ | AP$\uparrow$ | FPR$\downarrow$ |
>     | COCO | 69.41 | 9.46 | **90.54** | **1.98** |
>     | POC alt. | 68.8 | 11.4 | 87.4 | 3.1 |
>     | POC c. | 73.0 | **9.2** | 87.0 | 2.1 |
>     | CGAug (ours) | **76.56** | 10.17 | 85.70 | 7.16 |

---

### Official Review · Reviewer_UbMA · 2024-07-12

**Soundness:** 3
**Presentation:** 3
**Contribution:** 4
**Rating:** 5
**Confidence:** 4

**Summary:**

This paper aims to tackle both covariate-shift and semantic-shift in semantic segmentation. The idea is to use a generative augmentation method to produce coherent images that incorporate both anomaly objects and various covariate shifts at both image and object levels. The semantic segmentation model is then on the synthetic image for recalibrating uncertainty for semantic shifts and enhances the feature extractor to align features associated with domain shifts. The authors have conducted extensive experiments to show the effectiveness of the proposed method over the state-of-the-art methods.

**Strengths:**

1. The paper tackles an important problem in semantic segmentation and is well-motivated.

2. The solution is mostly reasonable and well-motivated. The paper is mostly well-written and organized.

3. The authors have compared with recent state-of-the-art OOD segmentation methods and demonstrated impressive performance.

**Weaknesses:**

1. The novelty of the proposed solution is somewhat limited. While the generative-based data augmentation appears reasonable, its novelty is not clearly articulated.

2. The proposed relative contrastive loss involves several hyperparameters, but it is unclear how to effectively tune these parameters in practice.

3. The necessity of the Two-Stage Noise-Aware Training is not clear. Why not use a single-stage training process that trains the uncertainty function and the feature extractor simultaneously?

4. What are conditioned label masks?

5. How can we determine if a pixel has a clean or incorrect label given the augmented images? Equation 6 seems to be used for assessing the accuracy of the pixel label, but why is this an effective strategy? How is α determined?

6. The proposed method involves many hyperparameters, which reduces its applicability in real-world scenarios.

**Questions:**

Please see weaknesses.

---

> ### Author Response · Authors · 2024-08-07
> **Rebuttal by Authors**
>
> Thank you for your time and constructive feedback. We discuss the novelty of our proposed generative-based augmentation in the general response. We address your other concerns below.
>
> # Weakness 2: Hyperparameters for Relative Contrastive Loss
>
> Our relative contrastive loss includes three terms, each with a margin value controlling the distance penalty limits. **These margins are set based on the average uncertainty scores from the training set.** Specifically, we compute the differences in uncertainty scores between unknown vs. original known data, unknown vs. augmented known data, original known vs. augmented known data, and set the differences as margins for these distance respectively.  A histogram of uncertainty scores is provided in the Figure 1 (c) attached PDF for reference.
>
> Moreover, our two-stage training framework first trains the uncertainty function based on the existing model, allowing this function to adapt to different scales. **This provides flexibility in parameter setting even without prior knowledge.** The experimental results, shown in Table 1 of the attached PDF, demonstrate the model's robustness across a wide range of hyperparameter variations. Specifically, maintaining the loss values within the same order of magnitude ensures that variations in parameters do not significantly affect the results.
>
> # Weakness 3: Two-Stage
>
> Thanks for pointing this out. The motivation and benefits of our two-stage training design are clarified below.
>
> - We observe that an initialized model with closed-world pre-training achieves a good feature representation while the initial uncertainty function, such as the energy score, is typically sub-optimal. Directly training the feature extractor with a sub-optimal uncertainty function risks disrupting the well-learned feature representations, which can harm known class segmentation and OOD detection (cf. Figure 4 (a)).
> - Our two-stage training approach addresses this challenge by first optimizing the uncertainty mapping head based on the current feature representations. This allows the subsequent fine-tuning of the feature extractor to be more effective and less disruptive.  Additionally, having a well-initialized uncertainty head before joint training helps minimize task competition between known class segmentation and OOD detection.
>
> - In Figure 4(a), we empirically compare our two-stage training with single-stage training, showing that our approach (noted as 'second stage') outperforms the 'single stage' by nearly 10 points in AP. With a closer look, our first-stage training, which involves training only the uncertainty function already significantly improves the baseline model's performance (from 45% to 85% in AP). This demonstrates the large performance gap between different uncertainty functions under the same feature extractor.
>
> # Weakness 4: Conditioned Label Mask
> The term "conditioned label masks" in the context of Line 208 refers to the label masks used to generate images. As explained in Eq. (1), these masks are created by cut-and-pasting the masks of novel objects onto the original training labels.
>
> # Weakness 5: Noise-aware training
> - **Sample Selection Mechanism**: We use the 'small loss' criterion to determine whether a pixel has a clean label, a simple and widely used technique in the noisy label learning literature [b1]. Specifically, we calculate and rank the cross-entropy loss for each pixel. Pixels with smaller losses are selected for backpropagation, while those with larger losses are ignored (cf. Eq. (5-6)). This is effective because during training, a model first learns simple and clean patterns before fitting noisy data [b2]; We will include explanations and relevant literature in the paper to aid understanding. A visualization of our sample selection results is illustrated in Figure 4 (b) of the paper.
>
> - **Determining the Selection Ratio α:** We determine α by visualizing the selection map of a small batch of data under several choices to ensure that visibly incorrect patterns are removed. To address the reviewer's concern, we conducted additional experiments with selection ratios ranging from 0.6 to 0.9, as detailed in Figure 1 (a) of the supplementary PDF. The results show that while including too many pixels (1.0) introduces noise, and including too few (0.6) removes useful regions, the model performance is stable within a wide range (0.7 to 0.9), demonstrating the robustness of the model to this hyperparameter.
>
> # Weakness 6: Hyperparameters & Applicability
>
> We have demonstrated robustness to loss margins and selection ratio in the previous response, supporting the practicality of our approach for real-world applications.
>
> [b1] Jiang, Lu, et al. Mentornet: Learning data-driven curriculum for very deep neural networks on corrupted labels. ICML, 2018.
> [b2] Arpit, Devansh, et al. A closer look at memorization in deep networks. ICML, 2017.

---

> > ### Comment · Reviewer_UbMA · 2024-08-12
> > **Thank you the clarifications.**
> >
> > I would like to thank the authors for the response. Since in the real-world applications of OOD segmentation, it is hard or impossible to tune the hyperparameters which somewhat limits the applicability of the method. But the overall, the problem is the interesting and the proposed method is reasonable. Therefore, I will maintain my original rating.

---

> ### Author Response · Authors · 2024-08-13
> **Thank you and Clarification on Hyperparameters**
>
> Thank you for your reply and for acknowledging our problem setting and proposed method.
>
> Regarding your concern about hyperparameter tuning, as detailed in our responses to weaknesses 2 and 5, the hyperparameters in our method, including loss margins and selection ratios, can be set directly based on training data statistics without the need for further tuning. Additionally, in the attached PDF, we demonstrate the robustness of our method across a wide range of these hyperparameters. Other normal training hyperparameters, like the learning rate, are tuned once per backbone and then kept consistent throughout experiments. We want to emphasize that our method does not require more hyperparameter tuning than previous approaches like RPL and Mask2Anomaly. Developing an OOD training strategy that eliminates the need for hyperparameter tuning is an interesting direction, and we will leave it to future work.

---

### Official Review · Reviewer_LCzB · 2024-07-12

**Soundness:** 3
**Presentation:** 3
**Contribution:** 2
**Rating:** 5
**Confidence:** 4

**Summary:**

This paper addresses semantic segmentation in the presence of domain and semantic shifts. To enhance model robustness, the authors propose using a generative model guided by ground truth labels to generate domain-shifted images and further inpaint random negative data. Due to potential noise in the generative process, an additional module is introduced to filter incorrectly filled-in image areas, removing pixels with the highest loss in the generated image.

The segmentation model is trained with a contrastive loss that promotes high uncertainty in negative samples and consistent uncertainty between original and domain-shifted areas.

Training proceeds in two stages. First, an uncertainty function is trained on top of a frozen pretrained semantic segmentation model. Subsequently, both the uncertainty function and the segmentation model are trained simultaneously.

**Strengths:**

1. The method is straightforward and can be integrated with both standard and mask-based segmentation models.
2. Some good qualitative results are presented.
3. The paper is clearly written.

**Weaknesses:**

1. The first contribution regarding the presence of domain and semantic shifts has already been addressed in previous works [a] [b].
2. There is limited novelty, as the use of generative models for data augmentation [8] has already been proposed.
3. SOTA results on SMIYC [c] [d] were omitted.

[a] Zendel et al., WildDash - Creating Hazard-Aware Benchmarks, ECCV 2018
[b] Bevandic et al., Simultaneous Semantic Segmentation and Outlier Detection in Presence of Domain Shift, GCPR 2019
[c] Grcic et al., On Advantages of Mask-level Recognition for Outlier-aware Segmentation, CVPRW 2023
[d] Nayal et al. RbA: Segmenting Unknown Regions Rejected by All, ICCV 2023

**Questions:**

1. How is sampling done for Eq. 4?

**Limitations:**

1. The method is limited by the pretraining of the generative model

---

> ### Author Rebuttal · Authors · 2024-08-07
>
> Thank you for the time and constructive feedback. We discuss the novelty of our generative-based data augmentation and additional comparison results on SMIYC in the general response. We address your other concerns below.
>
> ## Weakness 1: First Contribution
> We thank the reviewer for highlighting the works by Zendel et al. [a] and Bevandic et al. [b]. These studies indeed demonstrate the necessity and feasibility of handling both semantic segmentation under domain shifts and anomaly segmentation by proposing datasets and baselines. However, our work differs in problem setting and method design. Below, we detail these differences and will revise the contribution description and related work section to better highlight our contributions.
>
> - Our work builds upon these foundations by delving deeper into the core challenges of simultaneously enhancing model performance in both areas. Specifically, **we provide a novel analysis of the limitations of Domain Generalization techniques in identifying anomaly objects,** demonstrating the problem of 'over-generalization' for unknown regions. We also address the challenges faced by current state-of-the-art anomaly segmentation techniques, which often make errors in distinguishing between known objects with covariate shifts and real novel/anomaly objects (cf. Fig. 1). Furthermore, our findings indicate that simply combining techniques from both domains does not always yield optimal results for jointly handling domain shifts and semantic shifts. This is due to their focus on different levels (e.g., image-level domain shifts or object-level semantic shifts), leaving the challenge of distinguishing object-level domain shifts and semantic shifts unresolved (cf. Sec. 4.4).
> - We also note that while [a] and [b] are seminal works in domain shifts and anomaly segmentation, **their problem settings are still in early stages.** For example, WildDash [a] contains very few image-level anomaly samples, and while [b] introduces object-level anomaly samples via cut-and-paste, it is limited to animals and involves significant artifacts. More recent benchmarks, such as SegmentMeIfYouCan and MUAD, offer domain and semantic shifts that better reflect real-world scenarios. We have tested existing OOD detection and common domain generalization methods on these benchmarks, highlighting their limitations.
>
> Overall, we believe that our work takes a step further in addressing the challenges of jointly handling the two distribution shifts and filling a literature gap. We expect that our results will advocate for more research into developing algorithms that can improve both generalization and anomaly detection. We appreciate the reviewer's suggestion and will incorporate a discussion of these works in the related work section to better illustrate the contributions of our study.
>
> # Questions:
> For the calculation of our relative contrastive loss (Eq. 4), we randomly sample an equal number of pixels from the unknown-class set, original known-class set, and augmented known-class set, to calculate the contrastive losses between unknown and (original or augmented) known pixels. The third contrastive loss term is directly calculated for all paired pixels from original and augmented images.
>
> # Limitations:
> We discussed this limitation of our method in the conclusion. Similarly to what the reviewer mentioned here, the proposed augmentation strategy could be impacted by the quality of the generative model.

---

> ### Comment · Reviewer_LCzB · 2024-08-13
> **Answer to the rebuttal**
>
> I find the rebuttal to be thorough enough to increase my score to borderline accept.

---

> > ### Author Response · Authors · 2024-08-13
> > **Thank you**
> >
> > Thank you again for your time and valuable comments. We are pleased to have addressed your concerns and sincerely appreciate your positive feedback on our work.

---

### Author Rebuttal · Authors · 2024-08-07

We thank all reviewers for their time and constructive feedback. Below, we discuss shared concerns and reply to each reviewer with individual responses.

# The novelty of our Generative-based Augmentation

We address reviewers’ concerns about the novelty of our coherent generative-based data augmentation (CG-Aug). Although our approach is similar to [8], [a1], and [34] in using generative models to enhance training data, there are significant differences in both methodologies and effects, which we summarize as follows:

1. **Simultaneous Generation of Multiple Distribution Shifts:** Our method generates data with multiple distribution shifts, including both semantic-level shifts (eg. novel objects) and domain-level shifts, in a single generation process. By contrast, [8, 34] focuses solely on generating novel objects within the same domain, and [a1] employs separate frameworks for generating novel objects and domain shifts, making their process more complex and time-consuming for generating one image with multiple distribution shifts.
2. **Coherent Novel Object Generation via Semantic Mask-to-Image:** We generate unknown objects using a semantic mask-to-image generation process, which retains the global context of the image and ensures a more natural integration of novel objects. In previous methods, [34] uses a style transfer model to change the global style of cut-and-pasted OOD objects, but the OOD objects and the environment still remain distinctly different. [8] and [a1] use inpainting on cropped patches and then blend these patches into the original image, leading to inconsistencies and requiring additional post-processing to remove artifacts.

To evaluate the design of our generative-based augmentation, we compare with three variations: (1) **Semantic-Shift Only (SS):** Generate images with semantic shift using POC. (2) **DS or SS:** Create a mixed dataset with either domain shifts (DS) using our semantic-mask-to-image process or semantic shifts (SS) using POC. (3) **DS and SS:** First generate DS data, then inpaint unknown objects. The second and third methods can be seen as applying [a1] to our problem in two ways. Results in Table A1 show that: adding domain shift data significantly improves performance over semantic-shift-only data. Jointly generating DS and SS in one image yields better results than generating them separately. Our method, which generates both DS and SS in one step, achieves the best performance, ensuring more coherence without artifacts and outperforming the two-step approach.

Additionally, a visualization comparison of our method and POC is provided in Figure 3 of the main paper. A comparison with POC’s official results is included in the attached PDF, showing that our method outperforms POC on three out of four evaluated datasets. These results further demonstrate the superiority of our augmentation design.

Table A1: Ablation Study of our Coherent Generative-based Augmentation (CG_Aug).  Results are shown on the RoadAnomly dataset, using our training methods with the Mask2Former network.

| Data | AUROC | AP | FPR@TPR95 |
| --- | --- | --- | --- |
| SS | 95.43 | 83.66 | 10.33 |
| DS or SS | 95.90 | 87.64 | 9.28 |
| DS and SS  | 96.47 | 89.08 | 8.16 |
| CG-Aug (Ours) | 97.94 | 90.17 | 7.54 |

[a1] Loiseauet al. Reliability in Semantic Segmentation: Can We Use Synthetic Data? ArXiv 2023

# Additional Comparison on SMIYC Benchmark

We thank the reviewer for bringing up these recent works (RbA and M2F-EAM). Below, we discuss our concern about fairness by including them in our benchmark and present the results in Table A1.

**Concern on Benchmark Comparison Fairness:** The primary goal of our experimental design is to validate the efficacy of the proposed methods. To ensure fairness, we kept the training data and model architecture consistent with previous methods, particularly the recent works RPL and Mask2Anomaly.  In contrast, RbA and M2F-EAM utilize different resources and architectures. M2F-EAM, for example, leverages the Mapillary Vistas dataset, which is significantly larger and more diverse than Cityscapes. Additionally, both M2F-EAM and RbA use the Swin-Transformer backbone for the Mask2Former architecture, whereas we use ResNet-50 as in Mask2Anomaly. These differences can significantly impact performance, as evidenced by the ablation studies in both RbA and M2F-EAM papers.

**Results and Analysis:** We provide additional comparisons with RbA and M2F-EAM in Table A2, using their officially reported scores in their papers. Our method outperforms RbA on all evaluation metrics despite using a weaker backbone. When compared to M2F-EAM, our method shows superior performance on the RoadAnomaly and SMIYC-RO tracks.

Table A1. Complementary comparison on OOD segmentation task, comparing our method with RbA and M2F-EAM using their reported results. Note that new results for RbA are available on the benchmark website, but without additional context, a fair comparison cannot be ensured.
|  |  |  | RoadAnomaly |  | SMIYC-RA |  | SMIYC-RO |  |
| --- | --- | --- | --- | --- | --- | --- | --- | --- |
| Method | Backbone | Training Set | AP $\uparrow$ | FPR@95 $\downarrow$f | AP $\uparrow$ | FPR@95 $\downarrow$ | AP $\uparrow$ | FPR@95 $\downarrow$ |
| Mask2Anomaly (ICCV 23) | ResNet-50 | Cityscapes | 79.70 | 13.45 | 88.70 | 14.60 | 93.30 | 0.20 |
| RbA (ICCV 23) | Swin-B | Cityscapes | 85.42 | 6.92 | 90.9 | 11.6 | 91.8 | 0.5 |
| M2F-EAM (CVPRW 23) | Swin-L | Cityscapes + Mapillary Vistas | 69.4 | 7.7 | 93.8 | 4.1 | 92.9 | 0.5 |
| Ours | ResNet-50 | Cityscapes | 90.17 | 7.54 | 91.9 | 7.9 | 95.3 | 0.1 |

---

### Decision · Program_Chairs · 2024-09-25

**Decision:**

Accept (poster)

**Comment:**

This paper studies the problems of domain generalization and anomaly segmentation in tandem, showing that sometimes improving model performance in one direction is detrimental for performance on the other. They propose a new method to tackle both sources of distributional shifts at the same time.

Before the rebuttal, this work had a borderline rating (BR, BA, BA, WA). The main point of the most negative review was the novelty with respect to prior art. The Authors exhaustively commented on this, and the Reviewer found the answer satisfactory enough to increase their score to BA. After the rebuttal, all Reviewers are leaning towards accepting this paper (2 BA and 2 WA).

I agree with the Reviewers on the importance of the problem studied in this paper, and recommend acceptance. The Authors should revise the paper according to the feedback received by the Authors, and further clarify the concepts that were not clear to some of the Reviewers -- for example, the core differences with most related works.